# CARPRT: Class-Aware Prompt Reweighting for Pre-Trained Vision-Language Models

## Abstract

When using a pre-trained *vision-language model* (VLM) to classify an image, we often need to use the pre-trained VLM to compute a similarity score between the image and texts containing a semantic label, e.g., "a photo of a cat", where "a photo of a" is called a prompt and "cat" is the semantic label (a.k.a. a class in classification tasks). The existing studies have shown that the selection of prompts can significantly affect the scoring scheme between a given image and a semantic label, and they proposed a new score via using a *weighting vector* to reassemble scores regarding different prompts. However, these studies assume that all classes should share the *same* weighting vector. In this paper, we first empirically show that the existing approach is sub-optimal. We subsequently revisit the existing reweighting strategy from a probabilistic view and find an implicit assumption in prior work: the conditional independence of classes and weights, which often does not hold in practice. To cope with this problem, we propose *class-aware prompt reweighting* (CARPRT), a strategy designed to adjust the weighting vector for each class. CARPRT calculates the relevance scores for prompt-class pairs with respect to all images, and identifies the maximum score for each prompt-class pair. These maximum scores are then averaged across prompts for each class to estimate the class-specific weighting vectors, ensuring that prompts are optimally reweighted based on class-specific information. Our experiments demonstrate that CARPRT outperforms the existing reweighting strategy under the image classification tasks.

## 1 Introduction

Vision-language models (VLMs), such as CLIP (Radford et al., 2021), ALIGN (Jia et al., 2021), and LiT (Zhai et al., 2022), have transformed the way models understand visual content by leveraging information from both visual and textual modalities. VLMs can perform zero-shot classification tasks by encoding previously unseen class labels through text prompting (Radford et al., 2021), yet their performance remains highly sensitive to the quality of the prompts (Pham et al., 2023; Radford et al., 2021; Karmanov et al., 2024). These prompt templates, which are human-crafted textual structures, embed downstream class labels for specific tasks. However, relying on a single prompt often results in a lack of robust performance across different downstream tasks. For example, applying a prompt like "*a type of animal*" for EuroSAT (Helber et al., 2019)–satellite images for land-use classification–leads to poor accuracy due to lack of relevance between the prompt and the image. To address this limitation, prompt ensembling (Radford et al., 2021) averages the text embeddings of multiple prompts containing class names, forming a more reliable class-representative embedding, which in turn improving both accuracy and robustness across target tasks.

Prompt ensembling may yield suboptimal results when the number of prompts is too large and includes some that are unsuitable for the specific downstream tasks (Radford et al., 2021). There is a growing need for more automated approaches that can identify and weight the most effective prompts from a large prompt template pool without human intervention. Allingham et al. (2023) introduced a zero-shot prompt weighting method that assigns weights to each prompt template from the pool based on downstream data, achieving performance comparable to hand-crafted selections. However, this method assumes that the optimal weights are the same across different classes, which overlooks the diverse characteristics of different classes. This raises a question: *is it reasonable to apply the same prompt weights to all classes in a dataset*.

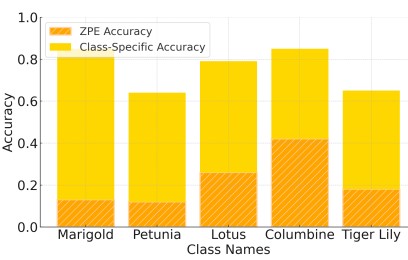 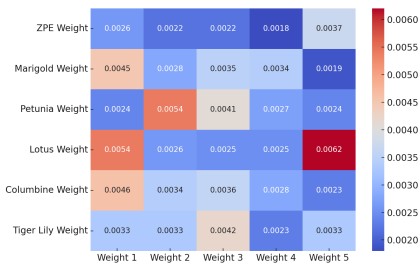

(a) Accuracy Differences between ZPE and Class-Specific Weights

(b) Weight Differences between ZPE and Class-Specific Weights

Figure 1: The optimal weights vary across different classes, even within the same downstream dataset. (a) On the Flower102 dataset, the five classes with the most pronounced accuracy discrepancies between zero-shot zero-shot prompt ensembling (ZPE) (Allingham et al., 2023) and class-specific weights are highlighted. (b) A heatmap visualizes the weight differences between ZPE and class-specific weights across five components. Red regions indicates larger weight differences, while blue regions indicate smaller or negligible differences, showing how the weights differ for each of the selected classes.

Intuitively, different classes and prompt templates vary in importance. For example, the prompt template *"This is a photo of a [label], a type of fruit"* is clearly more suitable when the [label] is *"strawberry,"* while *"This is a photo of a label, a type of animal"* is likely to provide more accurate information for labels like *"lamb."* To substantiate this intuition, we conducted proof-of-concept experiments on Flower102 (Nilsback & Zisserman, 2008), comparing the classification accuracy of class-specific weights against zero-shot prompt ensembling (ZPE) (Allingham et al., 2023). Class-specific weights were derived by applying ZPE separately for each class, tailoring the weights to the unique characteristics of each class. As Figure 1(a) shows, class-specific weights consistently yield higher classification accuracy, indicating that zero-shot prompt weights may not be universally optimal across different classes. Figure 1(b) further highlights significant variations in that the class-specific weights, suggesting that the optimal weights indeed differ across different classes.

Moreover, in Section 3, we present a probabilistic viewpoint based on Bayes' Theorem to understand weighted prompt ensembling in the zero-shot classification context, where the interactions of prompts, classes and images can be characterized with probabilistic tools. Our analysis uncovers a key implicit limitation in the class-shared-weighting (e.g., ZPE) method: the conditional independence assumption between the class and weights, which is not always satisfied in practice, resulting in limiting the expressivity of class-shared-weighted prompt ensembling schemes.

Inspired by our findings, in Section 4, we propose a *class-aware prompt reweighting* method called CARPRT in the zero-shot context[1], aiming to infer a unique weight vector for each class. Building upon CLIP (Radford et al., 2021), CARPRT computes the relevance score for each image and prompt-class pair by calculating the similarity between their respective text and image embeddings. The image is then assigned with a pseudo-class label based on the highest relevance score across all prompt-class pairs. Subsequently, these pseudo-labels are used to derive class-aware weights. The weight vector for each class is formed by selecting the highest score for each prompt-class pair, ensuring that the resulting weights prioritize the most relevant prompts for each class. This aligns the prompt weighting process with the unique characteristics of each class.

We verify the efficacy of CARPRT using the open-source CLIP model over multiple benchmarks, including ten fine-grained classification tasks, ImageNet (Russakovsky et al., 2015) and its variant test sets. Experimental results show that CARPRT outperforms existing prompt reweighting methods and achieves state-of-the-art performance in classification accuracy, highlighting the potential of class-awareness as a promising new direction in addressing the prompt reweighting problem.

## 2    PRELIMINARY AND RELATED WORKS

**CLIP for Zero-shot Image Classification**. CLIP (Radford et al., 2021) is a VLM that achieves visual-text alignment through large-scale contrastive pre-training. It consists of an image encoder

---

[1]We do not assume labelled to be available for estimating the weights, thus the class-specific weighting in the proof-of-concept experiment (Figure 1(a) and 1(b)) is no longer applicable.

$f : \mathcal{X} \to \mathcal{Z}$ and a text encoder $g : \mathcal{T} \to \mathcal{Z}$, where $\mathcal{X}$ and $\mathcal{T}$ represent the image and text spaces, respectively, and $\mathcal{Z}$ is a shared embedding space. The alignment is driven by maximizing the cosine similarity between the embeddings of matched image-text pairs while minimizing it for non-matched pairs, enabling CLIP to capture semantic correlations between visual and textual data.

This alignment is extended to various downstream vision tasks, such as zero-shot image classification. Consider a class space $\mathcal{Y} = \{y_1, \ldots, y_C\}$, each class $y_c$ is mapped to a text description $\boldsymbol{t}_c$ via a prompt template $p : \mathcal{Y} \to \mathcal{T}$, e.g., $\boldsymbol{t}_c =$ "A photo of $\{y_c\}$.". The text encoder $g(\cdot)$ embeds these descriptions into $\mathcal{Z}$, i.e., $\boldsymbol{z}_c^{\mathrm{T}} = g(\boldsymbol{t}_c)$, which results in $C$ class embeddings, denoted as

$$\begin{bmatrix} \boldsymbol{z}_1^{\mathrm{T}} & \boldsymbol{z}_2^{\mathrm{T}} & \cdots & \boldsymbol{z}_C^{\mathrm{T}} \end{bmatrix}^{\top}. \tag{1}$$

Then for image $\boldsymbol{x} \in \mathcal{X}$, CLIP predicts the label by selecting the class whose text embedding has the highest cosine similarity $\mathrm{sim}(\cdot, \cdot)$ with the image embedding $\boldsymbol{z}^{\mathrm{I}} = f(\boldsymbol{x})$, such that

$$\hat{y} = \arg\max_{c \in \{1, \ldots, C\}} \mathrm{sim}\left(\boldsymbol{z}^{\mathrm{I}}, \boldsymbol{z}_c^{\mathrm{T}}\right). \tag{2}$$

This enables zero-shot classification based on semantic alignment without task-specific fine-tuning. Yet, when a prompt template lacks task-specific relevance, the semantic inconsistency between the prompt template and the visual context can lead to misaligned class embeddings.

**Prompt Ensembling (PE)**. Radford et al. (2021) aim to address the issue above by prompt ensembling (PE) that leverages *multiple* prompt templates and computes their text representations, to improve the robustness of class embeddings. PE introduces $\mathbb{P} = \{p_i\}_{i=1}^n$ as a pool of prompt templates, where each $p_i$ maps class $y_c$ to a textual description $\boldsymbol{t}_{i,c} = p_i(y_c)$, such that each $\boldsymbol{t}_{i,c}$ provides a semantically diverse perspective for $y_c$. Then, the text encoder $g(\cdot)$ embeds all these $n$ descriptions for each class $y_c$, which eventually yields the class embeddings for $\forall y_c \in \mathcal{Y}$, such that

$$\begin{bmatrix} \boldsymbol{z}_1^{\mathrm{T}} \\ \vdots \\ \boldsymbol{z}_C^{\mathrm{T}} \end{bmatrix} = \frac{1}{n} \left( \begin{bmatrix} \boldsymbol{z}_{1,1}^{\mathrm{T}} & \boldsymbol{z}_{2,1}^{\mathrm{T}} & \cdots & \boldsymbol{z}_{n,1}^{\mathrm{T}} \\ \vdots & \vdots & \ddots & \vdots \\ \boldsymbol{z}_{1,C}^{\mathrm{T}} & \boldsymbol{z}_{2,C}^{\mathrm{T}} & \cdots & \boldsymbol{z}_{n,C}^{\mathrm{T}} \end{bmatrix} \cdot \begin{bmatrix} 1 \\ \vdots \\ 1 \end{bmatrix} \right). \tag{3}$$

In other words, for each $y_c$, its class embedding $\boldsymbol{z}_c^{\mathrm{T}}$ is obtained by *averaging* the embeddings $\boldsymbol{z}_{i,c}^{\mathrm{T}}$ derived from all prompts $p_i \in \mathbb{P}$. In doing so, PE reduces sensitivity to each individual suboptimal prompt by leveraging the collective semantic information captured by a diverse set of prompts.

**Weighted Prompt Ensembling**. Allingham et al. (2023) propose ZPE that extends PE to further mitigate the impact of task-irrelevant prompts. Instead of uniform weighting as in PE (Eq. 3), ZPE scores each prompt and assigns higher weights to prompts with higher task-relevance scores, by using an unlabeled downstream dataset $\mathbb{D} = \{\boldsymbol{x}_j\}_{j=1}^m$. Concretely, define $\boldsymbol{w} = [w_1, \ldots, w_n]^{\top}$ as the weights encoding the relevance of each prompt to the task. The weight for prompt $p_i \in \mathbb{P}$ is computed as $w_i = \sum_j \max_{c \in \{1, \ldots, C\}} \mathrm{sim}(\boldsymbol{z}_j^{\mathrm{I}}, \boldsymbol{z}_{i,y_c}^{\mathrm{T}})/m$, where $\boldsymbol{z}_j^{\mathrm{I}}$ is the image embedding of $\boldsymbol{x}_j$, and $\boldsymbol{z}_{i,y_c}^{\mathrm{T}}$ is the text embedding under prompt $p_i$ for class $y_c$. This quantifies $w_i$ as the average maximum similarity across all samples, between the image embeddings and the text embeddings of prompt $p_i$ over all classes. This leads to the class embeddings for $\forall y_c \in \mathcal{Y}$ as

$$\begin{bmatrix} \boldsymbol{z}_1^{\mathrm{T}} \\ \vdots \\ \boldsymbol{z}_C^{\mathrm{T}} \end{bmatrix} = \frac{1}{n} \left( \begin{bmatrix} \boldsymbol{z}_{1,1}^{\mathrm{T}} & \boldsymbol{z}_{2,1}^{\mathrm{T}} & \cdots & \boldsymbol{z}_{n,1}^{\mathrm{T}} \\ \vdots & \vdots & \ddots & \vdots \\ \boldsymbol{z}_{1,C}^{\mathrm{T}} & \boldsymbol{z}_{2,C}^{\mathrm{T}} & \cdots & \boldsymbol{z}_{n,C}^{\mathrm{T}} \end{bmatrix} \cdot \begin{bmatrix} w_1 \\ \vdots \\ w_n \end{bmatrix} \right). \tag{4}$$

This implies a *weighted-aggregation* of the embeddings $\boldsymbol{z}_{i,c}^{\mathrm{T}}$ obtained from all prompts $p_i \in \mathbb{P}$ with respect to $\boldsymbol{w}$, enabling ZPE to focus more on prompts better aligned with the current task.

**Limitations**. While ZPE offers improvements over *mean-aggregated* PE, it still assumes that the optimal weight for each prompt $p_i$ is *constant* across all classes and ignores a crucial factor as Figure 1 reveals: *the same prompt can contribute differently to classification depending on the class*. Mathematically, this calls for an expansion from the weight vector $\boldsymbol{w}$ to a matrix $\mathbf{W} = \{\mathbf{W}_c\}_{y_c \in \mathcal{Y}}$, where each class $y_c$ has its own set of weights $\mathbf{W}_c = [w_{1,c}, \ldots, w_{n,c}]^{\top}$, leading to

$$\begin{bmatrix} \boldsymbol{z}_1^{\mathrm{T}} \\ \vdots \\ \boldsymbol{z}_C^{\mathrm{T}} \end{bmatrix} = \frac{1}{n} \left( \begin{bmatrix} \boldsymbol{z}_{1,1}^{\mathrm{T}} & \boldsymbol{z}_{2,1}^{\mathrm{T}} & \cdots & \boldsymbol{z}_{n,1}^{\mathrm{T}} \\ \vdots & \vdots & \ddots & \vdots \\ \boldsymbol{z}_{1,C}^{\mathrm{T}} & \boldsymbol{z}_{2,C}^{\mathrm{T}} & \cdots & \boldsymbol{z}_{n,C}^{\mathrm{T}} \end{bmatrix} \cdot \begin{bmatrix} w_{1,1} & w_{1,2} & \cdots & w_{1,C} \\ \vdots & \vdots & \ddots & \vdots \\ w_{n,1} & w_{n,2} & \cdots & w_{n,C} \end{bmatrix} \right). \tag{5}$$

This reformulation captures the varying relevance of each prompt for different classes, which enables more flexible aggregation of prompted embeddings, motivating a principled reweighting scheme that better reflects such class specificity. Existing PE methods have largely overlooked this aspect, nor attempted to understand *why* class specificity is necessary to determine prompt relevance and *how* statistical tools help to address it. To bridge this gap, we next present a probabilistic framework that establishes a principled connection between *class-aware reweighting* and zero-shot classification.

# 3 UNDERSTANDING PROMPT REWEIGHTING: A PROBABILISTIC VIEWPOINT

Eq. 2 describes zero-shot classification with CLIP as predicting the label $\hat{y}^*$ given a query image $\boldsymbol{x}^*$. Equivalently, this task can be framed as modeling $\Pr(y^* \mid \boldsymbol{x}^*, \mathbb{P}, \mathbb{D})$, i.e., the conditional probability of label $y^*$ given input $\boldsymbol{x}^*$, the set of prompts $\mathbb{P}$, and the unlabeled dataset $\mathbb{D} = \{\boldsymbol{x}_j\}_{j=1}^m$, regardless of the specific PE strategy used (whether based on Eq. (3), Eq. (4), or Eq. (5)). To probe how *prompt reweighting* influences this task, we now analyze its probabilistic structure, particularly taking into account the effect of prompts and weights.

Let $\mathbf{W} \in \mathcal{W}$ be a weight matrix. We begin by marginalizing over the weight space $\mathcal{W}$ as

$$
\begin{aligned}
\Pr(y \mid \boldsymbol{x}, \mathbb{P}, \mathbb{D}) &= \int_{\mathcal{W}} \Pr(y \mid \boldsymbol{x}, \mathbb{P}, \mathbb{D}, \mathbf{W}) \Pr(\mathbf{W} \mid \boldsymbol{x}, \mathbb{P}, \mathbb{D}) \mathrm{d}\mathbf{W} \\
&= \int_{\mathcal{W}} \Pr(y \mid \boldsymbol{x}, \mathbb{P}, \mathbb{D}, \mathbf{W}) \Pr(\mathbf{W} \mid \mathbb{P}, \mathbb{D}) \mathrm{d}\mathbf{W},
\end{aligned}
\tag{6}
$$

where the second equation results from the conditional independence between query image $\boldsymbol{x}^*$ and weights $\mathbf{W}$, as $\mathbf{W}$ is not updated based on $\boldsymbol{x}^*$ in the zero-shot setting. This decomposition suggests two essential tasks in zero-shot classification: (i) accurately modeling the weights $\Pr(\mathbf{W} \mid \mathbb{P}, \mathbb{D})$ and (ii) utilizing $\Pr(y \mid \boldsymbol{x}, \mathbb{P}, \mathbb{D}, \mathbf{W})$ to predict the label given a specific weight configuration. As such, we will continue to explore *how further expansions can inform and align with practical implementations.*

## 3.1 MODELING $\Pr(\mathbf{W} \mid \mathbb{P}, \mathbb{D})$

Given $m$ i.i.d samples, we express this probability using the Bayes' rule as

$$
\Pr(\mathbf{W} \mid \mathbb{P}, \mathbb{D}) \propto \Pr(\mathbf{W} \mid \mathbb{P}) \Pr(\mathbb{D} \mid \mathbf{W}, \mathbb{P}) = \Pr(\mathbf{W} \mid \mathbb{P}) \prod_{j=1}^m \Pr(\boldsymbol{x}_j \mid \mathbf{W}, \mathbb{P}),
\tag{7}
$$

where $\Pr(\boldsymbol{x}_j \mid \mathbf{W}, \mathbb{P})$ is further marginalized over the possible classes for each sample $\boldsymbol{x}_j$, such that

$$
\Pr(\boldsymbol{x}_j \mid \mathbf{W}, \mathbb{P}) = \sum_{y_c \in \mathcal{Y}} \Pr(\boldsymbol{x}_j \mid y_c, \mathbf{W}, \mathbb{P}) \Pr(y_c \mid \mathbf{W}, \mathbb{P}).
\tag{8}
$$

This allows us to express the likelihood $\Pr(\boldsymbol{x}_j \mid y_c, \mathbf{W}, \mathbb{P})$ as a sum over all classes, weighted by their the conditional class probabilities $\Pr(y_c \mid \mathbf{W}, \mathbb{P})$.

**Modeling** $\Pr(y_c \mid \mathbf{W}, \mathbb{P})$. When domain knowledge is unavailable, a common choice for estimating the probability $\Pr(y_c \mid \mathbf{W}, \mathbb{P})$ is to assume a uniform prior over the classes. However, in zero-shot classification, we assume the unlabeled dataset, when used with the predictions from a pre-trained CLIP, can provide a reliable empirical estimate of the class prior distribution.

**Proposition 1.** *Let $\mathbb{D} = \{\boldsymbol{x}_j\}_{j=1}^m$ be an unlabeled dataset with unobserved classes $\mathcal{Y} = \{y_1, \ldots, y_C\}$, and $\Pr(y_c)$ be the true class probability for class $y_c$. Let $\hat{y}$ be the pseudo label produced by a pre-trained model. For sufficiently large $m$, the empirical class distribution $\hat{\Pr}(y_c \mid \mathbf{W}, \mathbb{P})$ converges to $\Pr(y_c)$. Specifically, for any $\epsilon > 0$, we have:* $\Pr\left\{|\hat{\Pr}(y_c \mid \mathbf{W}, \mathbb{P}) - \Pr(y_c)| \geq \epsilon\right\} \leq 2\exp\left(-2m\epsilon^2\right),$

**Remark 1.** *Proposition 1 implies that we can approximate true distributions by:*

$$
\hat{\Pr}(y_c \mid \mathbf{W}, \mathbb{P}) = \frac{n_c}{\sum_{y_{c'} \in \mathcal{Y}} n_{c'}}, \ \forall y_c \in \mathcal{Y},
\tag{9}
$$

*where $n_c = \sum_{j=1}^m \mathbb{1}_{\hat{y}_j = y_c}$ denotes the number of times $y_c$ was predicted over all samples in $\mathbb{D}$. These predictions $\hat{y}_j$ can be produced by following Eq. 2 in using CLIP.*

**Modeling** $\Pr(\boldsymbol{x}_j \mid y_c, \mathbf{W}, \mathbb{P})$. This likelihood term represents the probability of observing an image $\boldsymbol{x}_j \in \mathcal{X}$ given a class $y_c$, prompt set $\mathbb{P}$, and weight matrix $\mathbf{W}$. To characterize it, we use Energy-based Models (EBMs) (LeCun et al., 2006) capable of modeling complex high-dimensional distributions. EBMs allow us to express any probability distribution by defining an *unnormalized* energy function, with the normalization enforced by a partition function. The likelihood $p(\boldsymbol{x}_j \mid y_c, \mathbf{W}, \mathbb{P})$ can thus be formulated in the form of an EBM as

$$p(\boldsymbol{x}_j \mid y_c, \mathbf{W}, \mathbb{P}) = \frac{1}{Z(y_c, \mathbf{W}, \mathbb{P})} \exp\left\{ \mathrm{sim}(\boldsymbol{z}_j^{\mathrm{I}}, \boldsymbol{z}_c^{\mathrm{T}}) \right\}, \tag{10}$$

where $\boldsymbol{z}_j^{\mathrm{I}} = f(\boldsymbol{x}_j)$ and $\boldsymbol{z}_c^{\mathrm{T}} = g(p_i(y_c))$. In this way, $\mathrm{sim}(\boldsymbol{z}_j^{\mathrm{I}}, \boldsymbol{z}_c^{\mathrm{T}})$ can be seen as the negative of the energy function. In EBMs, configurations with higher similarity (and hence lower energy) are more likely. The partition function $Z(y_c, \mathbf{W}, \mathbb{P}) = \int_{\mathcal{X}} \exp(\mathrm{sim}(\boldsymbol{z}^{\mathrm{I}}, \boldsymbol{z}_c^{\mathrm{T}})) \mathrm{d}\boldsymbol{x}$ ensures that the likelihood is properly normalized across all possible images. While estimating the exact likelihood is intractable, in classification we are interested in the relative likelihoods of different classes for a given image.

**Lemma 1** (Relative Likelihood). *The likelihood of an image $\boldsymbol{x}$, given class $c$, prompt weights $\mathbf{W}$ and a prompt pool $\mathbb{P}$, following the EBM defined in Eq. (10), is proportional to*

$$\Pr(\boldsymbol{x}_j \mid y_c, \mathbf{W}, \mathbb{P}) \propto \exp\left\{ \mathrm{sim}(\boldsymbol{z}_j^{\mathrm{I}}, \boldsymbol{z}_c^{\mathrm{T}}) \right\} \propto \exp\left\{ \sum_{i=1}^{n} (w_{i,c}\, \boldsymbol{z}_{i,c}^{\mathrm{T}})^{\top} \cdot \boldsymbol{z}^{\mathrm{I}} \right\}, \tag{11}$$

*where $\boldsymbol{z}_j^{\mathrm{I}} = f(\boldsymbol{x}_j)$ and $\boldsymbol{z}_{i,c}^{\mathrm{T}} = g(p_i(y_c))$ are image embeddings of sample $\boldsymbol{x}_j$ and text embeddings of class $y_c$ under prompt $p_i$, respectively.*

**Class-aware Weighting Matters**. Lemma 1 (proof in Appendix E) results from a *general* formulation of PE. For each class $y_c$, $\mathrm{sim}(\boldsymbol{z}_j^{\mathrm{I}}, \boldsymbol{z}_c^{\mathrm{T}})$ ends up with a linear combination of image-class similarities over all $n$ prompt-class pairs. Each pair has a distinct embedding $\boldsymbol{z}_{i,c}^{\mathrm{T}}$ and weight $w_{i,c}$ reflects its contribution to the classification of $y_c$. Under this framework, ZPE (Allingham et al., 2023) is a *special* case with a conditional independence assumption, i.e., $w_{i,c} = w_i$ for all $y_c \in \mathcal{Y}$ given $\mathbf{W}$ and $\mathbb{P}$. While simplifying the model, this assumption constrains the range of likelihood functions that ZPE can represent. We now examine the representational limitations of such class-independent weighting schemes.

**Proposition 2.** *Let $\mathcal{X}$ be the image space, $\mathcal{Y}$ be the class space. Given a set of prompts $\mathbb{P}$, for any prompt weighting scheme $S$ (cf. Eqs. (3-5)), define the representable likelihood set $\mathcal{F}_S$ as:*

$$\mathcal{F}_S = \{ f : \mathcal{X} \times \mathcal{Y} \to \mathbb{R}_+ \mid \exists \mathbf{W} \in \mathcal{W}_S, \mathbb{P}, \text{ s.t. } f(\boldsymbol{x}, y_c) \propto \Pr(\boldsymbol{x} \mid y_c, \mathbf{W}, \mathbb{P}) \},$$

*where $\mathcal{W}_S$ is the weight space under the scheme $S$. Let $\mathcal{F}_{\mathrm{CI}}$ and $\mathcal{F}_{\mathrm{CA}}$ be the representable likelihood set induced from class-independent weighting (cf. Eq. (4)) and class-aware weighting (cf. Eq. (5)) schemes. Then, we have: $\exists f^* \in \mathcal{F}_{\mathrm{CA}}$ such that $\forall f_{\mathrm{CI}} \in \mathcal{F}_{\mathrm{CI}}, \exists \boldsymbol{x} \in \mathcal{X}, y_c \in \mathcal{Y}$ where $f^*(\boldsymbol{x}, y_c) \neq f_{\mathrm{CI}}(\boldsymbol{x}, y_c)$.*

**Remark 2.** *Proposition 3 shows that class-independent weighting (e.g., ZPE) cannot fully capture the variety of likelihood functions representable by class-aware weighting. This further indicates that prompt weights $w_{i,c}$ must be class-specific to ensure that each class benefits from the most relevant prompts, as determined by the visual-text similarity measured by CLIP.*

## 3.2 Modeling $\Pr(y \mid \boldsymbol{x}, \mathbb{P}, \mathbb{D}, \mathbf{W})$

Zero-shot classification is a *training-free* process, meaning that we cannot optimize prompt weights using standard learning methods. We approximate $\Pr(y^* | \boldsymbol{x}^*, \mathbb{P}, \mathbb{D}, \mathbf{W})$ with $\Pr(y^* | \boldsymbol{x}^*, \mathbb{P}, \hat{\mathbf{W}})$, where $\hat{\mathbf{W}}$ is considered a point estimate that captures information from $\mathbb{D}$, as we have discussed[2] in Eq. (9) and Eq. (11). Concretely, by considering each individual prompt from the prompt set $\mathbb{P}$, we have

$$\Pr(y^* | \boldsymbol{x}^*, \mathbb{P}, \hat{\mathbf{W}}) = \sum_{p_i \in \mathbb{P}} \Pr(y^* | \boldsymbol{x}^*, p_i, \hat{\mathbf{W}}) \propto \frac{\exp\left( \sum_{i=1}^{n} (w_{i,c}\, \boldsymbol{z}_{i,c}^{\mathrm{T}})^{\top} \cdot \boldsymbol{z}_*^{\mathrm{I}} \right)}{\sum_{c' \in 1, \dots, C} \exp\left( \sum_{i=1}^{n} (w_{i,c'}\, \boldsymbol{z}_{i,c'}^{\mathrm{T}})^{\top} \cdot \boldsymbol{z}_*^{\mathrm{I}} \right)}, \tag{12}$$

By now, we have framed CLIP-based zero-shot classification in a probabilistic framework (Eq. (6)), justified class-aware prompt reweighting (Propositions 1 and 3), and interpreted how class prediction for a query image can be performed (Eq. (12)) under this framework.

---

[2] The relevant content of weights prior $\Pr(\mathbf{W} | \mathbb{P})$ is deferred to Appendix D to keep the focus of the main text on class-aware reweighting, which is the central theme of this study.

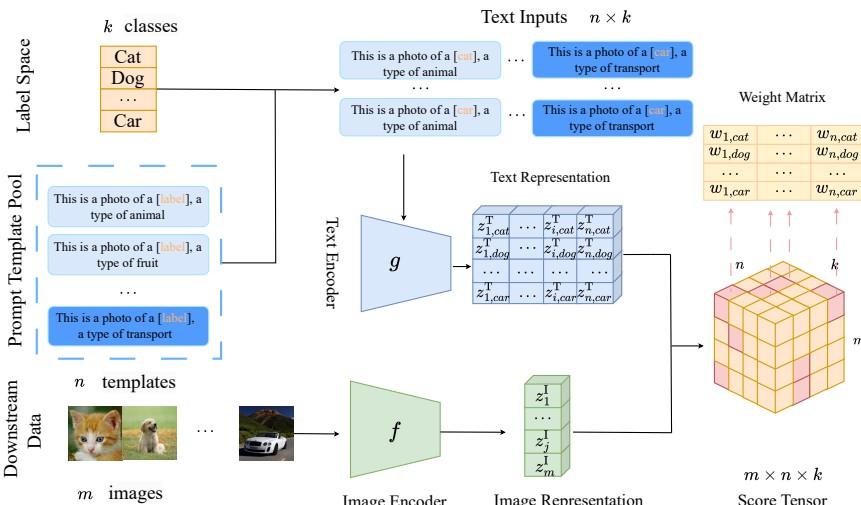

Figure 2: Overview of CARPRT. The model first generates the text input with the template pool $\mathbb{P}$ of prompt templates based on the label space $\mathcal{Y}$. These text inputs are then processed by a text encoder $g$ to generate text representations. Simultaneously, an image encoder $f$ generates representations for the downstream unlabeled images. The score tensor is generated by comparing the text and image representations, and these scores are then used to estimate the weight matrix $\mathbf{W}$, which adjusts the importance of each prompt template for each class.

# 4 CLASS-AWARE PROMPT REWEIGHTING FOR VLMS

We now introduce CARPRT, a minimalistic yet effective implementation that adheres to the key principles established in Section 3. CARPRT is designed to enhance zero-shot classification with CLIP by adaptively reweighting prompts according to their relevance to each class.

**Overview.** Given an unlabeled dataset $\mathbb{D} = \{\boldsymbol{x}_j\}_{j=1}^m$, an unknown class label space $\mathcal{Y} = \{y_1, \ldots, y_C\}$, a fixed prompt set $\mathbb{P} = \{p_i\}_{i=1}^n$, and a pre-trained CLIP model, define the weight matrix for all prompts in $\mathbb{P}$ as $\mathbf{W}$, such that $\mathbf{W} = \{\mathbf{W}_c\}_{y_c \in \mathcal{Y}}$, where each $\mathbf{W}_c = [w_{1,c}, \ldots, w_{n,c}]^\top$ lies on an $(n-1)$-dimensional simplex. The goal of CARPRT is to find a *class-aware weight matrix* $\mathbf{W}^* \in \mathbf{W}$ for classifying the images $\forall \boldsymbol{x} \in \mathbb{D}$ with CLIP, without using any true labels. As shown in Figure 2, CARPRT consists of two main stages: *Score Calculation* and *Weight Calculation*. The full process is described in Algorithm 4.1.

## 4.1 PROMPT RELEVANCE SCORE CALCULATION

As Eqs.(7-8) suggest, the key to estimating $\Pr(\mathbf{W}|\mathbb{P}, \mathbb{D})$ lies in individual likelihood $\Pr(\boldsymbol{x}_j|y_c, \mathbf{W}, \mathbb{P})$. According to Lemma 1, $\Pr(\boldsymbol{x}_j|y_c, \mathbf{W}, \mathbb{P})$ is fully consistent with the similarity score provided by CLIP. Thus, in the first stage, CARPRT calculates the similarity scores between each image embedding and each prompted-class embedding. Given an input image $\boldsymbol{x}_j \in \mathbb{D}$, the prompt template $p_i \in \mathbb{P}$ and the relevance score $s_{j,i,c}$ of $\boldsymbol{x}_j$ and $p_i$ belongs to the class $y_c \in \mathcal{Y}$ can be expressed by:

$$s_{j,i,c} = \text{sim}(\boldsymbol{z}_j^{\text{T}}, \boldsymbol{z}_{i,c}^{\text{T}}) \tag{13}$$

where $\boldsymbol{z}_j^{\text{I}} = f(\boldsymbol{x}_j)$ denotes the image embedding and $\boldsymbol{z}_{i,c}^{\text{T}} = g(p_i(y_c))$ refers the text embedding generated for class $y_c$ under prompt $p_i$. Eq. (13) provides a unnormalized estimate of $\Pr(\boldsymbol{x}_j|y_c, \mathbf{W}, \mathbb{P})$ and serves as the foundation for reweighting prompt-template combinations.

**Correcting Frequency Bias.** Before moving on to weight calculation, we take an additional debiasing step to correct the relevance scores. Frequency bias arises when the frequent concepts that appear in test images $\mathbb{D}$, but do not necessarily correspond to the target classes for prediction. Allingham et al. (2023) propose to mitigate this bias by subtracting expected scores derived from the pre-training dataset of CLIP. However, since neither the true pre-training dataset of CLIP, nor the open-sourced Laion-400M (Schuhmann et al., 2021) used as pre-training data in ZPE, are not publicly available at the time of this study, we adapt this approach for CARPRT by using test data instead.

---

**Algorithm 1** Class-Aware Prompt Reweighting (CARPRT)

---

**Input:** Pre-trained CLIP with image encoder $f$ and text encoder $g$, a prompt set $\mathbb{P}$, an unlabeled dataset $\mathbb{D}$, a candidate label space $\mathcal{Y}$ and the temperature parameter $\tau$ and the normalization scale $\lambda$.

**1: Generate** prompted-class texts $p_i(y_c), \forall p_i \in \mathbb{P}, \forall y_c \in \mathcal{Y}$;

**2: Encode** image embeddings $z_j^{\mathrm{I}} = f(x_j), \forall x_j \in \mathbb{D}$;

**3: Encode** text embeddings $z_{i,c}^{\mathrm{T}} = g(p_i(y_c)), \forall p_i \in \mathbb{P}, \forall y_c \in \mathcal{Y}$;

**4: Obtain** the relevance score set $\mathbb{S} = \{s_{j,i,c}\}_{j=1,i=1,c=1}^{m,n,C}$ by Eq. (13) ;

**5: Obtain** the normalized score by Eq. (14);

**6: Obtain** the pseudo-labels set: $\hat{\mathbb{Y}} = \{\hat{y}_{j,i}\}_{j=1,i=1}^{m,n}$;

**7: Derive** the weight matrix $\mathbf{W}^*$ by Eq. (15) and Eq. (16);

**Output:** a class-aware prompt weight matrix $\mathbf{W}^*$.

---

Specifically, CARPRT normalizes the relevance scores by subtracting the expected scores calculated over a subset of the test data. Let $\mathbb{S}$ denote the relevance scores computed with Eq. (13) from all test data. We set a sampling rate $\gamma$, which specifies the proportion of test data scores used in normalization. Then, the corrected relevance score for a sample $x_j$, prompt $p_i$ and the class $y_c$ is:

$$\bar{s}_{j,i,c} = s_{j,i,c} - \frac{1}{\gamma|\mathbb{S}|} \sum_{j'=1}^{\gamma|\mathbb{S}|} s_{j',i,c}. \tag{14}$$

In this way, the corrected scores better reflect the actual relative importance of prompts in relation to the specific target classes being predicted.

### 4.2 Class-aware Weight Calculation

In the second stage, CARPRT estimates the class-specific weights for each prompt-class combination based on the relevance scores $s_{j,i,c}$ computed in Eq. (13). These weights adjust the importance of each prompt template, ensuring that the most relevant prompts are emphasized for each class. The weight estimation process unfolds as follows.

First, we create a pseudo-label set $\hat{\mathbb{Y}} = \{\hat{y}_{j,i}\}_{j=1,i=1}^{m,n}$, where the pseudo-label $\hat{y}_{ji}$ for each sample $x_j$ under prompt $p_i$ is determined as the class $y_c$ that maximizes the relevance score $s_{j,i,c}$, *i.e.*, $\hat{y}_{ji} = \arg\max_{y_c \in \mathcal{Y}} s_{j,i,c}$. Then, we compute intermediate weight $w'_{i,c}$ for each prompt-class pair by aggregating the scores $s_{j,i,c}$ across all images $x_j$ predicted to belong to class $y_c$ under prompt $p_i$. This can be expressed as:

$$w'_{i,c} = \frac{\sum_{j=1}^{m} s_{j,i,c} \mathbb{1}_{\hat{y}_{ji}=y_c}}{\sum_j \mathbb{1}_{\hat{y}_{ji}=y_c}}. \tag{15}$$

Here, $\mathbb{1}_{\hat{y}_{ji}=y_c}$ is an indicator function that is 1 if $\hat{y}_{j,i} = y_c$, and 0 otherwise. This aligns with the empirical estimate of the class prior probabilities as indicated by Eq. (9). Afterward, a softmax normalization is applied to these intermediate weights to obtain the final weight $w^*_{i,c}$,

$$w^*_{i,c} = \frac{\exp\left(w'_{i,c}/\tau\right)}{\sum_c \exp\left(w'_{i,c}/\tau\right)}, \tag{16}$$

where $\tau$ is the temperature that controls the sharpness of the distribution. The use of softmax ensures the probabilistic validity of the weights for each class, i.e., $\sum_{i=1}^{n} w_{i,c} = 1$. By constructing $w^*_{i,c}$ in this manner, we integrate empirical class distributions into the reweighting scheme, ensuring that $w^*_{i,c}$ reflects both the relevance scores (Eq. (8)) and the estimated class priors (Eq. (9)), thus providing a principled inference approach to *class-aware prompt reweighting*.

## 5 Experiments

**Setup**. We evaluate the effectiveness of CARPRT on ten fine-grained classification benchmarks, include Caltech101, DTD, EuroSAT, Aircraft, Food101, Flowers102, Pets, Cars, Sun397 and UCF101. We include further evaluations on ImageNet along with its variant test sets: ImageNet-R, ImageNet-A, ImageNet-Sketch, and ImageNet-V2. See Appendix B for details of the datasets. We adhere to the established experimental protocol by (Zhou et al., 2022b). For prompt templates, we adopt

Table 1: Accuracy (%) comparison between baselines and our method % on various fine-grained classification datasets using CLIP-ViT-B/16 and CLIP-ResNet50 backbones. **Bold** value represents the highest accuracy on each column. Standard deviations are shown on the second row for ZPE and CARPRT.

| | Caltech101 | DTD | EuroSAT | Aircraft | Food101 | Flower102 | Pets | Cars | SUN397 | UCF101 | Average |
|---|---|---|---|---|---|---|---|---|---|---|---|
| **CLIP-ViT-B/16** | | | | | | | | | | | |
| Equal Weight | 92.50 | 46.88 | 51.86 | 21.49 | 85.34 | 64.21 | 79.46 | **65.21** | 64.92 | 67.41 | 63.93 |
| ZPE | 92.49 | 47.25 | 52.26 | 21.60 | 85.48 | 66.10 | 79.85 | 64.73 | 64.87 | 68.28 | 64.29 |
| | ±0.08 | ±0.63 | ±0.03 | ±0.28 | ±0.05 | ±0.06 | ±0.58 | ±0.07 | ±0.02 | ±0.17 | ±0.20 |
| CARPRT | **92.60** | **47.74** | **55.85** | **22.64** | **85.78** | **68.58** | **82.48** | 65.02 | **65.49** | **68.61** | **65.48** |
| | ±0.07 | ±0.68 | ±0.03 | ±0.24 | ±0.05 | ±0.11 | ±0.49 | ±0.07 | ±0.01 | ±0.16 | ±0.19 |
| **CLIP-ResNet50** | | | | | | | | | | | |
| Equal Weight | 86.41 | 41.69 | 30.34 | 16.05 | 75.53 | 56.95 | 75.98 | 55.74 | 59.32 | 60.06 | 55.81 |
| ZPE | 85.83 | 41.94 | **30.95** | 16.24 | 75.61 | 56.67 | 74.79 | 55.67 | 59.21 | 61.06 | 55.80 |
| | ±0.06 | ±0.39 | ±0.03 | ±0.17 | ±0.07 | ±0.07 | ±0.63 | ±0.03 | ±0.01 | ±0.25 | ±0.17 |
| CARPRT | **86.63** | **42.66** | 30.88 | **16.29** | **76.08** | **60.01** | **76.94** | **55.75** | **59.90** | **61.87** | **56.70** |
| | ±0.06 | ±0.42 | ±0.03 | ±0.15 | ±0.06 | ±0.09 | ±0.51 | ±0.03 | ±0.01 | ±0.16 | ±0.16 |

the pool of 247 prompts as used by Allingham et al. (2023). Additionally, we conduct ablation and hyper-parameter studies to analyze the behavior of our method. Details regarding baselines, implementations and our anonymous code repository are in Appendix C.

We note that the results of ZPE reported in Tables 1 and Tables 2 differ from those in the original study Allingham et al. (2023). This discrepancy arises primarily because ZPE used the LAION-400M (Schuhmann et al., 2021) dataset as pre-trained data for normalization to correct for frequency biases. The dataset is no longer publicly accessible, precluding us from evaluating ZPE with the same normalization process. Also, the ZPE implementation used a batch size of 5,000, a configuration we could not replicate due to computational limitations.

## 5.1 RESULTS ON FINE GRAINED DATASETS

**Overall Comparison**. Following the data split used by Zhou et al. (2022b), we evaluate the model performances on 10 fine-grained classification tasks, as shown in Table 1. Our method outperforms the baselines in most tasks. For example, on EuroSAT, we achieve a significant improvement of 3.59% over ZPE using the CLIP-ViT-B/16 backbone. Similarly, on Flower102, we record a 2.48% increase in accuracy. On average, using the CLIP-ViT-B/16 backbone, our method yields an improvement of 1.19% over ZPE across all datasets, demonstrating the overall efficacy of our method.

**Scalability**. We further evaluate the scalability of CARPRT using the CLIP-ResNet50 backbone. As shown in the lower half of Table 1, our method consistently outperforms ZPE. For example, on EuroSAT, CARPRT achieves an improvement of 2.93% over ZPE, and a 0.92% increase is observed on average. These results demonstrate that our approach generalizes well across different backbone.

**Quality of Prompts Matters**. The performance gain is less obvious or there is a performance gap on some datasets, such as Cars and Aircraft. This may be attributed to the lower quality of the pseudo-labels generated by CLIP for these datasets, which directly impacts the performance of our method, as it relies heavily on pseudo-label accuracy. Additionally, the template pool used for these datasets was manually crafted for the entire dataset, rather than being tailored to individual classes(Radford et al., 2021; Zhai et al., 2022). This may result in reduced class differentiation, leading to smaller variations in the class-specific weights. Nonetheless, our method exhibits superior performance overall, particularly when class-specific prompt relevance is critical.

## 5.2 RESULTS ON IMAGENET AND ITS VARIANTS DATASETS RESULT

**Overall Comparison**. We also evaluate the performance of our method across ImageNet and its variant datasets (ImageNet-A, ImageNet-R, ImageNet-Sketch, and ImageNet-V2), as shown in Table 2. Our method outperforms the baselines, though the gains are modest. For CLIP-ViT-B/16, CARPRT achieves 60.70 %, compared to 60.51 % accuracy on average. When using the CLIP-ResNet50 backbone, our method also shows an improvement in accuracy, i.e., 53.81% (ZPE) vs 53.72% (CARPRT) on average.

Table 2: Accuracy (%) comparison between baselines and our method % on ImageNet and its variants using CLIP-ViT-B/16 and CLIP-ResNet50 backbones. **Bold** value represents the highest accuracy on each column. Standard deviations are shown on the second row for ZPE and CARPRT.

|  | ImageNet | -A | -R | -Sketch | -V2 | Equal Weight |
|---|---|---|---|---|---|---|
| CLIP-ViT-B/16 | | | | | | |
| Average | 67.59 | 49.35 | 77.33 | 46.92 | 61.37 | 60.51 |
| ZPE | 67.42 | **49.84** | 77.28 | 47.14 | 61.11 | 60.58 |
|  | ±0.01 | ±0.12 | ±0.03 | ±0.02 | ±0.11 | ±0.06 |
| CARPRT | **67.81** | 49.12 | **77.48** | **47.53** | **61.58** | **60.70** |
|  | ±0.01 | ±0.07 | ±0.04 | ±0.02 | ±0.09 | ±0.05 |
| CLIP-ResNet50 | | | | | | |
| Equal Weight | 59.12 | 46.25 | 69.05 | 39.05 | 54.05 | 53.50 |
| ZPE | 59.78 | **46.37** | 69.27 | 39.14 | 54.07 | 53.72 |
|  | ±0.01 | ±0.08 | ±0.01 | ±0.07 | ±0.09 | ±0.06 |
| CARPRT | **59.98** | 46.19 | **69.38** | **39.25** | **54.26** | **53.81** |
|  | ±0.02 | ±0.09 | ±0.01 | ±0.04 | ±0.03 | ±0.06 |

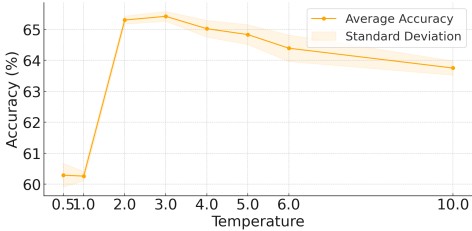

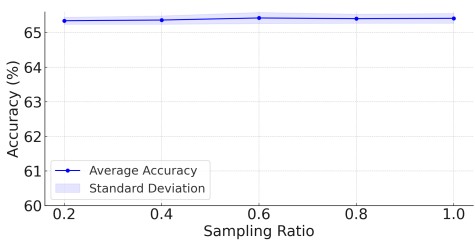

(a) Temperature Hyper-Parameter Analysis      (b) Sampling Ratio Hyper-Parameter Analysis

Figure 3: Hyper-Parameter Analysis on Fine-Grained Datasets.The shaded area represents the standard deviation. Subfigure (a) illustrates the variation of accuracy with temperature adjustments. Subfigure (b) demonstrates the stability of accuracy across different sampling ratios.

**Analysis of Incremental Improvements in ImageNet Performance**. The improvements on ImageNet and its variants datasets are smaller compared to those observed on the fine-grained datasets, for the following reasons. First, frequency bias is likely more pronounced in ImageNet and its variants. Given our use of a relatively small batch size of 512 and the exclusion of larger datasets such as LAION-400M for debiasing, the skewed class distribution may have negatively impacted the results. Second, the quality of the template pool plays a crucial role in model performance. According to (Allingham et al., 2023), the template pool was constructed by combining templates from 10 fine-grained datasets and 6 ImageNet and its variants datasets. Fine-grained datasets benefit more from the pool, as they can exploit class-specific templates. In contrast, the more diverse categories in ImageNet and its variants find less relevant information in the fine-grained templates, deriving less benefit from these templates. This mismatch reduces the overall effectiveness of our method on ImageNet datasets as it relies on the information provided by the templates. These limitations suggest that addressing frequency bias and improving the relevance of templates for broader datasets could lead to more substantial performance improvements in future iterations of CARPRT.

**Hyper-Parameter Sensitivity**. Figure 3 illustrates two key hyper-parameters on the performance of CARPRT: the temperature parameter $\tau$ and the sampling ratio $\gamma$ used during score normalization, spanning 10 fine-grained datasets.

The temperature parameter $\tau$ controls the sharpness of the weight distribution across prompt templates, directly affecting how strongly the model emphasizes relevant prompts. As illustrated in Figure 3(a), the accuracy peaks at a temperature of 3.0 and maintains relatively high stability across a range of values, with a slight decline as the temperature further increases. The under-performance observed at $\tau = 0.5$ can be explained by the way temperature affects the distribution of weights across prompt templates. A lower temperature, such as 0.5, sharpens the focus on the most probable prompts

Table 3: Accuracy (%) comparison with uniform weight and class-aware weights across 10 fine-grained datasets. **Bold** value represents the highest accuracy on each column.

| | Caltech101 | DTD | EuroSAT | Aircraft | Food101 | Flower102 | Pets | Cars | SUN397 | UCF101 | Average |
|---|---|---|---|---|---|---|---|---|---|---|---|
| Uniform | 92.48 | 47.13 | 54.17 | 21.64 | 85.35 | 67.12 | 81.04 | 65.09 | 64.99 | 68.03 | 64.46 |
| Class-Aware | **92.60** | **47.49** | **55.85** | **22.53** | **85.78** | **68.41** | **82.39** | 65.02 | **65.49** | **68.61** | **65.42** |

but also reduce the distribution's spread. Given the use of 247 prompt templates, even those with marginal relevance collectively contribute to enhancing model robustness and generalization through an ensemble effect. This effect allows the model to capture a wider range of information cues. When the temperature is set too low, the model becomes overly concentrated on dominant prompts, potentially overlooking broader information captured by less dominant prompts. The sampling ratio $\gamma$ governs the fraction of the test data used for estimating the expected scores during bias correction. As Figure 3(b) shows, accuracy consistent performance across different sampling levels, implying robustness with varying data availability. Details regarding hyper-parameter analysis are in Appendixe H.

**Class-aware Weight Matters**. We further examine the "uniformity" of weight vectors. We test with a configuration where the class-aware weights derived by CARPRT are collapsed into a uniform weight vector as $w_i^{\mathrm{u}} = \frac{1}{C} \sum_c w_{i,c}$. This aggregation assesses whether the complexity of class-specific weights is necessary or if a simplified, averaged representation can achieve similar performance. It helps determine if the merits of class-aware weights lies in their specificity or if a general representation suffices for certain tasks. Results in Table 3 show that models with class-aware weights consistently outperform those using uniform weights. The improvement is most evident in datasets with pronounced class-specific traits, such as EuroSAT, Flower102, and Pets, where accuracy increases significantly. These results highlight the importance of adapting weights to class-specific traits, as uniform weights may hinder CLIP to exploit the semantic differences across classes.

## 6 DISCUSSION AND FUTURE OUTLOOK

**Related Works**. Prompts play a vital role in adapting pre-trained VLMs to downstream tasks. Alongside prompt reweighting, prompt tuning methods such as CoOp (Zhou et al., 2022b), CoCoOp (Zhou et al., 2022a) and MaPLe (Khattak et al., 2023a), have also been actively explored, focusing on optimizing task-specific prompts. In contrast with these *training-based* methods, CARPRT focuses on better utilizing existing prompts in a *training-free* manner. Test-time adaptation, on the other hand, *updates* feature statistics (Wang et al., 2021) or *fine-tune* the prompts (Shu et al., 2022) to adapt to each test sample during inference, whereas CARPRT leaves the model and prompts *unchanged* but reweights existing prompts based on their relevance to the test data. This makes CARPRT orthogonal to prompt tuning and test-time adaptation. We report additional results of combining CARPRT with test-time adaptation in Appendix G. See Appendix A for detailed discussion of related works.

**Summary**. This study focused on prompt ensembling and confirmed that class-aware prompt reweighting is not only beneficial but essential for improving the efficacy of VLMs across a variety of downstream classification tasks. By moving beyond uniform weighting, we showed that adapting weights to better reflect the class-specific characteristics leads to measurable gains in classification accuracy. We hope this study encourages further exploration of integrating class-awareness with other VLM adaptation techniques to enhance across a wider range of applications.

**Future Work**. As per previously discussed limitations, two specific avenues for future work stand out: First, refining the estimation of class-specific weights could enhance class-aware prompt reweighting. Existing methods often rely on top-1 pseudo-labels, which may fail in complex tasks with multiple plausible labels. A promising alternative is to explore top-k pseudo-labels, which have been shown to yield higher accuracy in applications such as CLIP, where top-level classifications provide a richer set of potential alignments between visual inputs and textual prompts. Second, the efficacy of CARPRT is tied to the quality and diversity of the prompt template pool, which however is overlooked by current methods. Future work may focus on cost-effective strategies for creating and evaluating diverse and representative prompts. This could involve developing metrics that assess how well prompts capture the distinctive characteristics of different classes and methods that amplify inter-class differences could improve model performance in differentiating closely related categories.

ACKNOWLEDGMENTS

The authors would like to thank all reviewers for their valuable and constructive comments in advance.

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

## A  Detailed Related Works

**Prompt tuning methods.**  Prompt tuning adapts a pre-trained model by introducing learnable embeddings, known as prompt tokens, at the input stage. These tokens can be either text prompts or visual prompts, enabling flexible adjustments to the model's input interface to better address specific tasks. CoOp was the first to apply prompt tuning in CLIP, optimizing learnable prompts within its textual branch for few-shot image recognition (Zhou et al., 2022b). Addressing CoOp's limitations, CoCoOp introduces conditionally generated prompts based on visual features to enhance generalization performance (Zhou et al., 2022a). Further, MaPLe advances a multi-modal approach, applying prompt tuning simultaneously within the vision and textual branches to facilitate better transfer capabilities (Khattak et al., 2023a). Building upon MaPle, PromptSRC employs a strategy that enhances textual prompt learning by utilizing descriptive text generated by large language models (LLMs), such as GPT-4 (Khattak et al., 2023b). However, this approach requires updating learnable input variables in the text or image inputs, leading to additional computational resources and labeled downstream data, even if only few-shot data is used. Since our problem setting differs from that of tuning methods, we do not include such approaches as baselines in our experiments with CARPRT.

**Test-time Adaptation**  The *test-time adaptation* (TTA) problem is aim to adapts models adapts models to testing downstream data (Ganin et al., 2016; Long et al., 2015; Zhang et al., 2022). TTA methods can be diveded into two types: the training-based method and the training-free method. Training-based methods typically involve updating model weights or fine-tuning prompts based on test data (Zhang et al., 2022). TTA methods, such as TENT, adapt models by optimizing for test-time objectives like entropy minimization, adjusting the model's batch normalization statistics to align with the test distribution (Wang et al., 2021). CoTTA have explored contrastive learning to preserve feature space alignment, making TTA effective for CLIP-like models (Chen et al., 2022). TPT addresses the challenge in vision-language models by fine-tuning a learnable prompt for each individual test sample (Shu et al., 2022). DiffTPT extends this approach by utilizing pre-trained diffusion models to increase the diversity of test data samples used in TPT, enhancing the effectiveness of test-time prompt tuning (Feng et al., 2023).

On the other hand, non-training methods rely on adjusting normalization statistics or augmenting test samples without changing model parameters (Li et al., 2016; Karmanov et al., 2024). Since the problem setting of non-training TTA methods, which only require unlabeled test data and do not involve additional training, aligns with the CARPRT setup, we analyze the non-training TTA methods in comparison to CARPRT in Appendix G.

## B  Datasets

**Fine-grained datasets.**  Following Zhou et al. (2022b), we evaluate our method in 10 different fine-grained datasets. Caltech101 (Fei-Fei et al., 2004): A dataset containing images of objects belonging to 101 different categories, commonly used for object recognition tasks; DTD (Cimpoi et al., 2014): A texture dataset containing images categorized by describable texture attributes such as "bumpy" or "scaly"; EuroSAT (Helber et al., 2019): A dataset for land use and land cover classification, consisting of satellite images across 10 classes such as residential, forest, and river; Aircraft (Maji et al., 2013): A fine-grained dataset containing aircraft images, used for recognizing and classifying different airplane models; Food101 (Bossard et al., 2014): A large dataset containing 101 food categories, designed for image recognition tasks in the food domain; Flower102 (Nilsback & Zisserman, 2008): A fine-grained flower classification dataset with 102 different types of flowers, used for challenging image recognition tasks; Oxford Pets (Parkhi et al., 2012): A dataset consisting of images of 37 pet breeds, used for fine-grained image classification tasks; Cars196 (Krause et al., 2013): A fine-grained dataset for car model classification, with 196 car classes focused on vehicle recognition; SUN397 (Xiao et al., 2010): A large-scale scene recognition dataset with 397 scene categories, covering a wide variety of environments; UCF101 (Khurram, 2012): A dataset for action recognition in videos, containing 101 human action categories captured in realistic video scenarios.

**ImageNet and its Variant datasets.**  Following (Allingham et al., 2023), we also evaluate our method in ImageNet and the following variants of the ImageNet dataset: ImageNet (Russakovsky et al., 2015): A large-scale dataset for image classification, containing over 14 million labeled images

Table 4: Details for the datasets in our experiments.

| Dataset | Classes | Test Size |
|---|---|---|
| ImageNet | 1000 | 50,000 |
| ImageNet-R | 200 | 30,000 |
| ImageNet-A | 200 | 6862 |
| ImageNet-Sketch | 1000 | 50,889 |
| ImageNet-V2 | 1000 | 10,000 |
| Caltech101 | 100 | 2465 |
| DTD | 47 | 1692 |
| EuroSat | 10 | 8100 |
| Aircraft | 100 | 3333 |
| Food101 | 101 | 30,300 |
| Flowers102 | 102 | 2463 |
| Oxford Pets | 37 | 3669 |
| Cars196 | 196 | 8041 |
| Sun397 | 397 | 19,850 |
| UCF101 | 101 | 3783 |

across 1,000 object categories; ImageNet-A (Hendrycks et al., 2021b): A curated subset of ImageNet consisting of challenging adversarial images that fool standard models, designed to test the robustness of image classifiers; ImageNet-R (Hendrycks et al., 2021a): A dataset containing renditions of ImageNet objects in diverse artistic forms, such as paintings, cartoons, and sculptures, used to assess model performance on non-photorealistic images; ImageNet-Sketch (Wang et al., 2019): A sketch-based dataset derived from ImageNet, used to evaluate model robustness and generalization to line drawings of objects; ImageNet-V2 (Recht et al., 2019): A reproduction of the original ImageNet test set collected under similar conditions, used to measure model generalization to a newly collected version of the dataset.

## C   DETAILS REGARDING EXPERIMENTS

**Implementation Details.** We implement all methods using PyTorch 1.7.1 and Python 3.7.6, and conduct all experiments on a single NVIDIA A100 Tensor Core GPU. Our vision-language model is built on the architecture and pretrained weights from OpenCLIP (Radford et al., 2021). The code for our experiments is available at https://anonymous.4open.science/r/CARPRT-8402/ provided for reproducibility.

**Hyper-parameter Settings.** We set fixed hyper-parameters for the different datasets. The temperature $\tau$ is set to 3.0 for fine-grained datasets and 5.1 for ImageNet (Russakovsky et al., 2015) and its variants. The sampling ratio $\gamma$ is consistently set to 0.8 for both types of datasets, and the batch size is fixed at 512 for all experiments.

## D   DISCUSSION OF $\Pr(\mathbf{W}|\mathbb{P})$

We extend the discussion of the proposed probabilistic interpretation (Section 3) to the weights prior $\Pr(\mathbf{W}|\mathbb{P})$. In the current zero-shot classification scenario addressed by CARPRT, there is no optimization-based process for "estimating" the weights, and as such, the weight prior $\Pr(\mathbf{W}|\mathbb{P})$ does not play a role in the methodology. Nevertheless, our probabilistic framework is flexible enough to accommodate more general trainable settings, such as active learning and few-shot estimation, where the probabilistic formulation becomes particularly beneficial. In these cases, a discussion of the weight prior would provide valuable insights and contribute to a more complete understanding of the framework's advantages.

Suppose there is a label space $\mathcal{Y}$ with size $|\mathcal{Y}| = C$. Let $\mathbb{P} = \{p_i\}_{i=1}^n$ be a pool of $n$ independent prompt templates. Let $\mathbf{W} = \{\mathbf{W}_c\}_{c=1}^C$ be our weight matrix. Recall that $\mathbf{W}_c \in \Delta^{n-1}$ is the $(n-1)$-dimensional probability simplex, representing the weights for class $y_c$ across all prompts.

We consider three choices of priors: uniform prior, global Dirichlet prior, and class-specific Dirichlet priors.

**Uniform Prior**. The uniform prior assumes all valid weight configurations are equally likely a priori.

$$p(\mathbf{W}|\mathbb{P}) = \begin{cases} \frac{1}{|\mathcal{W}|} & \text{if } \mathbf{W} \in \mathcal{W} \\ 0 & \text{otherwise} \end{cases}$$

where $\mathcal{W} = \{\mathbf{W} \in \mathbb{R}^{n \times C} : W_c \in \Delta^{n-1} \text{ for all } c \in \{1, ..., C\}\}$.

The uniform prior is the easiest setup to implement and does not introduce bias towards any particular weight configuration. However, the uniform prior does not leverage any prior knowledge about the prompts, which is prone to overfitting with limited data (when adapted to trainable setting).

**Global Dirichlet Prior**. This defines a single Dirichlet distribution over all weights, treating them as a single vector.

$$p(\mathbf{W}|\mathbb{P}) = \text{Dir}(\text{vec}(\mathbf{W})|\alpha_1, ..., \alpha_{nC})$$

where $\text{vec}(\mathbf{W})$ is the vectorization of $\mathbf{W}$, and $\alpha_i > 0$ are concentration parameters of the Dirichlet distribution.

Compared to uniform prior, Dirichlet prior can encode varying degrees of certainty about different weights. Moreover, it is conjugate to multinomial likelihood, allowing for closed-form posterior updates for certain model setup. This can also align with ZPE-like class-shared-weighting strategies. However, it ignores the class structure and treats all weights as part of a single distribution, potentially missing class-specific patterns.

**Class-specific Dirichlet Prior**. This strategy sets an independent Dirichlet distribution for each class's weight, and stacks a product of $C$ classes' Dirichlet distributions.

$$p(\mathbf{W}|\mathbb{P}) = \prod_{c=1}^{C} \text{Dir}(\mathbf{W}_c|\alpha_{c,1}, ..., \alpha_{c,n})$$

where $\alpha_{c,i} > 0$ are class and prompt-specific contenration parameters.

Currently, this setup best suits our class-aware prompt reweighting mechanism, as it allows for different prior beliefs about weight distributions for each class, class-specific modeling. Compared with global Dirichlet, it reduces dimensionality - each Dirichlet distribution is over $n$ parameters, not $n \times C$ anymore. More importantly, it aligns with the per-class simplex constraint of the weight space.

**Entropy Analysis**. Different prior choices lead to different entropy results. The uniform prior has an associated entropy as

$$H[p(\mathbf{W}|\mathbb{P})]_{\text{uniform}} = \log|\mathcal{W}|,$$

where $|\mathcal{W}|$ is the volume of the weight space.

As for global Dirichlet prior, we have

$$H[p(\mathbf{W}|\mathbb{P})] = \log B(\alpha) + (\alpha_0 - nC)\psi(\alpha_0) - \sum_{i=1}^{nC} (\alpha_i - 1)\psi(\alpha_i),$$

where $B(\cdot)$ is the multivariate beta function, and $\psi(\cdot)$ is the digamma function.

The entropy for class-specific Dirichlet priors is

$$H[p(\mathbf{W}|\mathbb{P})] = \sum_{c=1}^{C} (\log B(\alpha_c) + (\alpha_{c,0} - n)\psi(\alpha_{c,0}) - \sum_{i=1}^{n} (\alpha_{c,i} - 1)\psi(\alpha_{c,i})),$$

where $\alpha_c = (\alpha_{c,1}, ..., \alpha_{c,n})$ and $\alpha_{c,0} = \sum_{i=1}^{n} \alpha_{c,i}$ for each class $c$.

When we are setting the equal concentration parameters, such that $\alpha_i = \alpha$ for all $i$ in the global Dirichlet, and $\alpha_{c,i} = \alpha$ for all $c, i$ in the class-specific Dirichlets, and let $\alpha = 1$, the uniform prior has the highest entropy (uninformative), while the class-specific Dirichlets having the lowest entropy. This is because the class-specific Dirichlets with $\alpha = 1$ are equivalent to independent uniform distributions over smaller simplices, further concentrating the probability.

# E  DETAILED PROOFS

**Lemma 2** (Relative Likelihood *cf.* Lemma 1)**.** *The likelihood of an image $\boldsymbol{x}$, given class $c$, prompt weights $\mathbf{W}$ and a prompt pool $\mathbb{P}$, following the EBM defined in Eq. (10), is proportional to:*

$$\Pr(\boldsymbol{x}_j \mid y_c, \mathbf{W}, \mathbb{P}) \propto \exp\left\{sim(\boldsymbol{z}_j^{\mathrm{I}}, \boldsymbol{z}_c^{\mathrm{T}})\right\} \propto \exp\left\{\sum_{i=1}^{n}(w_{i,c}\,\boldsymbol{z}_{i,c}^{\mathrm{T}})^{\top} \cdot \boldsymbol{z}^{\mathrm{I}}\right\}, \tag{17}$$

*where $\boldsymbol{z}_j^{\mathrm{I}} = f(\boldsymbol{x}_j)$ and $\boldsymbol{z}_{i,c}^{\mathrm{T}} = g(p_i(y_c))$ are image embeddings of sample $\boldsymbol{x}_j$ and text embeddings of class $y_c$ under prompt $p_i$, respectively.*

*Proof.* **Similarity as Negative Energy**. As with (LeCun et al., 2006), a general form of EBMs is given by $P_\theta(x) = \exp(-\beta E_\theta(x))/Z(\theta)$, which enables us to define unnormalized energy function with a partition function for normalization. Therefore, in our zero-shot classification context, we define the energy function with respect to the score function of the CLIP.

$$E(\boldsymbol{x}_j, y_c, \mathbf{W}, \mathbb{P}) = \mathrm{sim}(\boldsymbol{z}_j^{\mathrm{I}}, \boldsymbol{x}_c^{\mathrm{T}})$$

This score function measures the compatibility between the image embedding $\boldsymbol{z}_j^{\mathrm{I}}$ and the text embedding embedding $\boldsymbol{x}_c^{\mathrm{T}}$ of class $y_c$. higher compatibility corresponds to lower energy, aligning with the EBM principle that more likely configurations (of model) have lower energy.

**Intractable Partition Function**. Computing the partition function is intractable since we need to marginalize over the image space. However, what we care about is the relative relation between $\Pr(\boldsymbol{x}_j \mid y_c, \mathbf{W}, \mathbb{P})$ and $\Pr(\boldsymbol{x}_j \mid y_{c'}, \mathbf{W}, \mathbb{P})$, we can safely drop off the partition function in our relative likelihood.

**Similarity Computation**. Consider a general linear combination of similarities for a prompt ensemble:

$$\mathrm{sim}(\boldsymbol{z}^{\mathrm{I}}, \boldsymbol{z}_c^{\mathrm{T}}) = h_c\left(\{\mathrm{sim}(\boldsymbol{z}^{\mathrm{I}}, \boldsymbol{z}_{i,c}^{\mathrm{T}})\}_{i=1}^{n}\right)$$

$$h_c(\{s_i\}_{i=1}^{n}) = \sum_{i=1}^{n} \alpha_{i,c} s_i + \beta_c$$

where $h_c : \mathbb{R}^d \to \mathbb{R}$ is a function that linearly combines the similarities over all prompts $p_i \in \mathbb{P}$ for a specific class $y_c$. $\alpha_{i,c} \in \mathbb{R}$ and $\beta_c \in \mathbb{R}$ are weights and bias terms. Substituting $s_i = \mathrm{sim}(\boldsymbol{z}^{\mathrm{I}}, \boldsymbol{z}_{i,c}^{\mathrm{T}}) = \boldsymbol{z}_{i,c}^{\mathrm{T}\top} \cdot \boldsymbol{z}^{\mathrm{I}}$, we get:

$$\mathrm{sim}(\boldsymbol{z}_j^{\mathrm{I}}, \boldsymbol{z}_{i,c}^{\mathrm{T}}) = \sum_{i=1}^{n} \alpha_{i,c}(\boldsymbol{z}_{i,c}^{\mathrm{T}})^{\top} \cdot \boldsymbol{z}_j^{\mathrm{I}} + \beta_c$$

We can then absorb the bias term $\beta_c$ into the exponential function,

$$\Pr(\boldsymbol{x}_j \mid y_c, \mathbf{W}, \mathbb{P}) \propto \exp(\mathrm{sim}(\boldsymbol{z}_j^{\mathrm{I}}, \boldsymbol{z}_{i,c}^{\mathrm{T}}))$$

$$= \exp(\sum_{i=1}^{n} \alpha_{i,c}(\boldsymbol{z}_{i,c}^{\mathrm{T}})^{\top} \cdot \boldsymbol{z}_j^{\mathrm{I}} + \beta_c)$$

$$= \exp(\beta_c)\exp(\sum_{i=1}^{n} \alpha_{i,c}(\boldsymbol{z}_{i,c}^{\mathrm{T}})^{\top} \cdot \boldsymbol{z}_j^{\mathrm{I}})$$

$$\propto \exp(\sum_{i=1}^{n} (\alpha_{i,c}\boldsymbol{z}_{i,c}^{\mathrm{T}})^{\top} \cdot \boldsymbol{z}_j^{\mathrm{I}}).$$

By setting $w_{i,c} = \alpha_{i,c}$, we arrive at the formulation in Lemma 1. $\square$

**Proposition 3** (*cf.* Proposition 3)**.** *Let $\mathcal{X}$ be the image space, $\mathcal{Y}$ be the class space. Given a set of prompts $\mathbb{P}$, for any prompt weighting scheme $S$ (cf. Eqs. (3-5)), define the representable likelihood set $\mathcal{F}_S$ as:*

$$\mathcal{F}_S = \{f : \mathcal{X} \times \mathcal{Y} \to \mathbb{R}_+ \mid \exists \mathbf{W} \in \mathcal{W}_S, \mathbb{P}, \ s.t. \ f(\boldsymbol{x}, y_c) \propto \Pr(\boldsymbol{x} \mid y_c, \mathbf{W}, \mathbb{P})\},$$

*where $\mathcal{W}_S$ is the weight space under the scheme $S$. Let $\mathcal{F}_{CI}$ and $\mathcal{F}_{CA}$ be the representable likelihood set induced from class-independent weighting (cf. Eq. (4)) and class-aware weighting (cf. Eq. (5)) schemes. Then, we have: $\exists f^* \in \mathcal{F}_{CA}$ such that $\forall f_{CI} \in \mathcal{F}_{CI}, \exists \boldsymbol{x} \in \mathcal{X}, y_c \in \mathcal{Y}$ where $f^*(\boldsymbol{x}, y_c) \neq f_{CI}(\boldsymbol{x}, y_c)$.*

*Proof.* We prove this by constructing a specific function in $\mathcal{F}_{CA}$ and showing it cannot be represented by any function in $\mathcal{F}_{CI}$. For simplicity, we consider a **toy** setting with three classes $\mathcal{Y} = \{y_1, y_2, y_3\}$ and two prompts $\mathbb{P} = \{p_1, p_2\}$. For any $\boldsymbol{x} \in \mathcal{X}$, the function under class-aware weighting for $\forall\, y_c \in \{y_1, y_2, y_3\}$ takes the form:

$$f^*(\boldsymbol{x}, y_c) = \sum_{i=1}^{|\mathbb{P}|} w_{i,c} \Pr(\boldsymbol{x} \mid y_c, p_i)$$
$$= w_{1,c} \Pr(\boldsymbol{x} \mid y_c, p_1) + w_{2,c} \Pr(\boldsymbol{x} \mid y_c, p_2).$$

where $w_{i,j} \in \mathbb{R}_+$ are class-aware weights for prompt $i$ and class $j$. For ease of notation, we denote the prompt-conditional likelihood by $a_{i,c} \triangleq \Pr(\boldsymbol{x} \mid y_c, p_i)$. This way $f^* \in \mathcal{F}_{CA}$ can be expressed as

$$f^*(\boldsymbol{x}, y_1) = w_{1,1} a_{1,1} + w_{2,1} a_{2,1}$$
$$f^*(\boldsymbol{x}, y_2) = w_{1,2} a_{1,2} + w_{2,2} a_{2,2}$$
$$f^*(\boldsymbol{x}, y_3) = w_{1,3} a_{1,3} + w_{2,3} a_{2,3}$$

We then consider a specific instance[3] of this function by choosing:

$$w_{1,1} = 2, \quad w_{2,1} = 1$$
$$w_{1,2} = 1, \quad w_{2,2} = 2$$
$$w_{1,3} = 3, \quad w_{2,3} = 3$$

This leads to
$$f^*(\boldsymbol{x}, y_1) = 2a_{1,1} + a_{2,1}$$
$$f^*(\boldsymbol{x}, y_2) = a_{1,2} + 2a_{2,2}$$
$$f^*(\boldsymbol{x}, y_3) = 3a_{1,3} + 3a_{2,3}$$

Now, suppose for contradiction that $\exists f_{CI} \in \mathcal{F}_{CI}$ such that $f^* = f_{CI}$. By definition of $\mathcal{F}_{CI}$, $f_{CI}$ takes the form $f_{CI}(\boldsymbol{x}, y_c) = w_1 a_{1,c} + w_2 a_{2,c}$, where $w_1, w_2 \in \mathbb{R}_+$ are class-independent weights.

If $f^* = f_{CI}$, then for all classes $y_c \in \{y_1, y_2, y_3\}$, we must have the following equations to hold simultaneously:
$$2a_{1,1} + a_{2,1} = w_1 a_{1,1} + w_2 a_{2,1} \quad \text{(for } y_1)$$
$$a_{1,2} + a_{2,2} = w_1 a_{1,2} + w_2 a_{2,2} \quad \text{(for } y_2)$$
$$3a_{1,3} + 3a_{2,3} = w_1 a_{1,3} + w_2 a_{2,3} \quad \text{(for } y_3)$$

From these equations, we can deduce that
$$w_1 = 2 \text{ and } w_2 = 1 \text{ must hold for any } a_{1,1}, a_{2,1} > 0 \quad \text{(for } y_1)$$
$$w_1 = 1 \text{ and } w_2 = 2 \text{ must hold for any } a_{1,2}, a_{2,2} > 0 \quad \text{(for } y_2)$$
$$w_1 = 3 \text{ and } w_2 = 3 \text{ must hold for any } a_{1,3}, a_{2,3} > 0 \quad \text{(for } y_1)$$

Thus, we need $w_1 = 2$ for $y_1$ while $w_1 = 1$ for $y_2$, immediately leading to a contradiction as $w_1$ cannot simultaneously equal 1 and 2.

Therefore, no class-independent weighting scheme can represent the function $f^*$ we constructed. We have proven that $\exists f^* \in \mathcal{F}_{CA}$ such that $\forall f_{CI} \in \mathcal{F}_{CI}$, $\exists \boldsymbol{x} \in \mathcal{X}, y_c \in gY$ where $f^*(\boldsymbol{x}, y_c) \neq f_{CI}(\boldsymbol{x}, y_c)$. $\qquad\square$

# F  CONNECTING CARPRT FORMULATION WITH THE PROBABILISTIC FRAMEWORK

We now detail the correspondence between the CARPRT formulation (Section 4) and the probabilistic framework established in Section 3.

Concretely, the practical implementation Eqs. (13-16) align with Eqs.(7-11) in the following manner.

---

[3] unnormalized weights, just for illustration

Table 5: Accuracy (%) comparison between our method and baselines combing to TDA method using CLIP-ViT-B/16 and CLIP-ResNet50 backbones. **Bold** value represents the highest accuracy on each column.

| | Caltech101 | DTD | EuroSAT | Aircraft | Food101 | Flower102 | Pets | Cars | SUN397 | UCF101 | Average |
|---|---|---|---|---|---|---|---|---|---|---|---|
| CLIP-ViT-B/16 | | | | | | | | | | | |
| Human Select | **94.24** | 47.40 | 58.00 | 23.91 | 86.14 | **71.42** | **88.63** | 67.28 | 67.62 | 70.66 | 67.53 |
| Equal Weight | 93.18 | 46.75 | 60.60 | 23.37 | 86.04 | 65.61 | 84.21 | 67.44 | 66.41 | 71.48 | 66.51 |
| ZPE | 93.49 | 47.02 | 62.48 | 23.09 | 86.21 | 68.10 | 84.12 | 67.23 | 66.98 | 71.23 | 67.00 |
| CARPRT | 94.02 | **48.52** | **63.95** | **24.05** | **86.50** | 70.36 | 84.50 | **67.83** | **68.06** | **71.85** | **67.96** |
| CLIP-ResNet50 | | | | | | | | | | | |
| Human Select | 91.42 | 41.00 | 56.97 | **20.55** | 83.34 | **62.75** | **83.62** | 64.14 | 65.86 | 68.52 | 63.82 |
| Equal Weight | **92.03** | 41.77 | 54.56 | 19.77 | 83.41 | 62.50 | 80.65 | 63.55 | 64.14 | 68.80 | 63.12 |
| ZPE | 91.67 | 41.89 | 56.78 | 19.84 | 83.21 | 56.67 | 81.66 | 63.43 | 64.87 | 68.72 | 63.45 |
| CARPRT | 91.75 | **42.71** | **57.65** | 19.98 | **83.61** | 62.66 | 81.38 | **65.98** | 65.98 | 68.65 | 63.76 |

**Score Calculation**. Eq. (13) implements the likelihood term $\Pr(\boldsymbol{x}_j|y_c, W, \mathbb{P})$ from Eq. (11) by defining $s_{j,i,c} = \frac{\exp(a_{j,i,c}/\lambda)}{\sum_{y \in \mathcal{Y}} \exp(a_{j,i,c}/\lambda)}$. This formulation aligns with the EBM in Eq. (11) by using cosine similarity $a_{j,i,c}$ as the negative energy term and normalizing through softmax to obtain proper probabilities.

**Weight Calculation**. Eqs. (15-16) correspond to estimating $\Pr(W|\mathbb{P}, \mathbb{D})$ from Eq. (8) through a two-step process. Eq. (15) first obtains the pseudo-labels for samples as the empirical estimates $\hat{\Pr}(y_c|W, \mathbb{P})$ (i.e., Eq. (9)). It then estimates intermediate weights by aggregating scores across pseudo-labeled samples by multiplying the scores $\Pr(\boldsymbol{x}_j|y_c, W, \mathbb{P})$ (i.e., $s_{j,i,c}$) with $\hat{\Pr}(y_c|W, \mathbb{P})$. Eq. (16) applies softmax to ensure the resulting weights form a valid probability distribution over prompts for each class, which satisfies the simplex constraint implied by our probabilistic framework.

# G COMBINING CARPRT AND TEST-TIME ADAPTATION METHOD

Our method aligns more closely with the training-free TTA method as it operates without training, making it computationally efficient. TDA is a state-of-the-art, training-free test-time adaptation (TTA) method for CLIP that enables efficient and effective adaptation of vision-language models without backpropagation (Karmanov et al., 2024).

Our approach is not in conflict with TDA but is orthogonal to it. While TDA uses a human-selected prompt pool for each task, our method can serve as a complementary module that replaces this human selection pool, providing an alternative way of selecting prompts without requiring human intervention. This allows our method to work alongside TDA, enhancing the adaptability of vision-language models in a more automated manner. We conduct the experiment to compare the performance of our method with several baselines, including the human-selected prompts, the equal weight prompt selection, an ZPE, all combined with the TDA method. The results are evaluated using both CLIP-ViT-B/16 and CLIP-ResNet50 backbones across ten fine-grained datasets, as shown in Table 5.

From the result, we can observe that our method outperforms the other baselines in several datasets, achieving the highest average accuracy of 67.96% for CLIP-ViT-B/16 and 63.76% for CLIP-ResNet50. Specifically, for datasets like EuroSAT, Food101, and Flower102, our method shows significant improvements over the human-selected and ZPE baselines. These improvements demonstrate that our approach effectively enhances the performance of TTA methods, by offering a more efficient prompt selection strategy. However, there are cases where it falls short compared to human-selected prompts. This may be caused by the limited diversity and smaller size of the template pool, where automatic reweighting methods may not perform as well as direct human selection. However, the automated approach significantly reduces the human labor cost. This experiment demonstrates the promising future of our method—not only in prompt reweighting but also as a technique that can be integrated into other vision-language model (VLM) transfer learning approaches. The ability to automatically adjust prompts in a computationally efficient manner paves the way for broader applications and adaptability in various VLM-based tasks.

**Posterior Update with TTA.** When prompt weights can be updated continuously, such as in TTA settings, different priors (e.g., uniform, global Dirichlet, or class-specific Dirichlet) define initial beliefs about weight distributions before observing test data. In the TTA scenario, test data arrives as a stream: $\{\boldsymbol{x}^{(0)}, \ldots, \boldsymbol{x}^{(t)}, \boldsymbol{x}^{(t+1)}, \ldots\}$. Based on Eq. (8), we have a general form of posterior

$$p(\mathbf{W}|\boldsymbol{x}^{(t)}, \mathbb{P}) \propto p(\boldsymbol{x}^{(t)}|\mathbf{W}, \mathbb{P})p(\mathbf{W}|\mathbb{P}),$$

where $p(W|\mathbb{P})$ is the prior, $p(\boldsymbol{x}^{(t)}|\mathbf{W}, \mathbb{P})$ is the likelihood from test data, and $p(W|\boldsymbol{x}^{(t)}, \mathbb{P})$ is the posterior that guides weight updates sample-by-sample. The posterior updating process follows:

For first test sample $\boldsymbol{x}^{(0)}$:

$$\text{Prior}: p(\mathbf{W}|\mathbb{P})$$
$$\text{Likelihood}: p(\boldsymbol{x}^{(0)}|\mathbf{W}, \mathbb{P})$$
$$\text{Posterior}: p(W|\boldsymbol{x}^{(0)}, \mathbb{P}) \propto p(\boldsymbol{x}^{(0)}|\mathbf{W}, \mathbb{P})p(\mathbf{W}|\mathbb{P})$$

Then, as we observe the second test sample $\boldsymbol{x}^{(1)}$, we have

$$\text{Prior}: p(\mathbf{W}|\boldsymbol{x}^{(0)}, \mathbb{P}) \quad \text{(previous posterior)}$$
$$\text{Likelihood}: p(\boldsymbol{x}^{(1)}|\mathbf{W}, \mathbb{P})$$
$$\text{Posterior}: p(\mathbf{W}|\boldsymbol{x}^{(0)}, \boldsymbol{x}^{(1)}, \mathbb{P}) \propto p(\boldsymbol{x}^{(1)}|\mathbf{W}, \mathbb{P})p(\mathbf{W}|\boldsymbol{x}^{(0)}, \mathbb{P})$$

This leads to the sequential update scheme, formulated as

$$p(\mathbf{W}|\boldsymbol{x}^{(0)}, \ldots, \boldsymbol{x}^{(t)}, \mathbb{P}) \propto p(\boldsymbol{x}^{(t)}|\mathbf{W}, \mathbb{P})p(\mathbf{W}|\boldsymbol{x}^{(0)}, \ldots, \boldsymbol{x}^{(t-1)}, \mathbb{P})$$

Thus, in TTA settings, these priors can be (1) initialized based on initial test samples; and (2) updated sequentially as new test samples arrive.

More specifically, choosing different prior distributions would lead to different updating computations.

*Uniform Prior.* Recall the uniform prior is defined as

$$p(W|\mathbb{P}) = \begin{cases} \frac{1}{|\mathcal{W}|} & \text{if } W \in \mathcal{W} \\ 0 & \text{otherwise} \end{cases}$$

By taking log to both LHS and RHS, we will have

$$\log p(\mathbf{W}|\mathbb{P}) = \begin{cases} -\log |\mathcal{W}| & \text{if } \mathbf{W} \in \mathcal{W} \\ -\infty & \text{otherwise} \end{cases}$$

which then leads to the log posterior to be expressed as

$$\log p(\mathbf{W}|\boldsymbol{x}^{(t)}, \mathbb{P}) \propto -\log |\mathcal{W}| + \log \sum_{y_c \in \mathcal{Y}} p(\boldsymbol{x}^{(t)}|y_c, \mathbf{W}, \mathbb{P})p(y_c|\mathbf{W}, \mathbb{P})$$

$$= -\log |\mathcal{W}| + \log \sum_{y_c \in \mathcal{Y}} \exp \left( \sum_{i=1}^{n} \left( w_{i,c} \boldsymbol{z}_{i,c}^{\mathrm{T}} \right)^{\top} \cdot \boldsymbol{z}^{\mathrm{I}} \right) \cdot \frac{\mathbb{1}_{\hat{y}_{ji}=y_c}}{\sum_{j'} \mathbb{1}_{\hat{y}_{j'i}=y_c}}$$

*Global Dirichlet Prior.* The global Dirichlet prior treats all weights across classes as a single vector:

$$p(W|\mathbb{P}) = \text{Dir}(\text{vec}(W)|\alpha_1, \ldots, \alpha_{nC})$$

where $\text{vec}(\mathbf{W}) \in \mathbb{R}^{nC}$ is the vectorization of weight matrix W (here we denote $C = |\mathcal{Y}|$ as the cardinality of label space) Similarly, we will have the log prior and posterior as

$$\log p(\mathbf{W}|\mathbb{P}) = \log \text{Dir}(\text{vec}(\mathbf{W})|\alpha_1, \ldots, \alpha_{nC})$$

$$= \log \Gamma(\alpha_0) - \sum_{k=1}^{nC} \log \Gamma(\alpha_k) + \sum_{k=1}^{nC} (\alpha_k - 1) \log w_k \quad (\alpha_0 = \sum_{k=1}^{nC} \alpha_k)$$

$$= \log \Gamma(\sum_{k=1}^{nC} \alpha_k) - \sum_{c=1}^{C} \sum_{i=1}^{n} \log \Gamma(\alpha_{(c-1)n+i}) + \sum_{c=1}^{C} \sum_{i=1}^{n} (\alpha_{(c-1)n+i} - 1) \log w_{i,c}$$

and

$$\log p(\mathbf{W}|\boldsymbol{x}^{(t)}, \mathbb{P}) \propto \log p(\mathbf{W}|\mathbb{P}) + \log p(\boldsymbol{x}^{(t)}|\mathbf{W}, \mathbb{P}) - \log p(\boldsymbol{x}^{(t)}|\mathbb{P})$$

$$= \log \Gamma(\alpha_0) - \sum_{k=1}^{nC} \log \Gamma(\alpha_k) + \sum_{c=1}^{C} \sum_{i=1}^{n} (\alpha_{(c-1)n+i} - 1) \log w_{i,c}$$

$$+ \log \sum_{y_c \in \mathcal{Y}} p(x|y_c, \mathbf{W}, \mathbb{P}) p(y_c|\mathbf{W}, \mathbb{P})$$

$$= \log \Gamma(\alpha_0) - \sum_{k=1}^{nC} \log \Gamma(\alpha_k) + \sum_{c=1}^{C} \sum_{i=1}^{n} (\alpha_{(c-1)n+i} - 1) \log w_{i,c}$$

$$+ \log \sum_{y_c \in \mathcal{Y}} \exp \left( \sum_{i=1}^{n} \left( w_{i,c} \boldsymbol{z}_{i,c}^{\mathrm{T}} \right)^{\top} \cdot \boldsymbol{z}^{\mathrm{I}} \right) \cdot \frac{\mathbb{1}_{\hat{y}_{ji} = y_c}}{\sum_{j'} \mathbb{1}_{\hat{y}_{j'i} = y_c}}$$

*Class-specific Dirichlet Prior.* We again start from the prior definition

$$p(W|\mathbb{P}) = \prod_{c=1}^{C} \mathrm{Dir}(W_c|\alpha_{c,1}, ..., \alpha_{c,n})$$

then turn into the log prior and posterior

$$\log p(\mathbf{W}|\mathbb{P}) = \sum_{c=1}^{C} \log \mathrm{Dir}(W_c|\alpha_{c,1}, ..., \alpha_{c,n})$$

$$= \sum_{c=1}^{C} \left[ \log \Gamma(\alpha_{c,0}) - \sum_{i=1}^{n} \log \Gamma(\alpha_{c,i}) + \sum_{i=1}^{n} (\alpha_{c,i} - 1) \log w_{i,c} \right] \quad \left( \alpha_{c,0} = \sum_{i=1}^{n} \alpha_{c,i} \right)$$

and log posterior

$$\log p(\mathbf{W}|\boldsymbol{x}^{(t)}, \mathbb{P}) = \sum_{c=1}^{C} \left[ \log \Gamma(\alpha_{c,0}) - \sum_{i=1}^{n} \log \Gamma(\alpha_{c,i}) + \sum_{i=1}^{n} (\alpha_{c,i} - 1) \log w_{i,c} \right]$$

$$+ \log \sum_{y_c \in \mathcal{Y}} p(x|y_c, \mathbf{W}, \mathbb{P}) p(y_c|\mathbf{W}, \mathbb{P})$$

$$= \sum_{c=1}^{C} \left[ \log \Gamma(\alpha_{c,0}) - \sum_{i=1}^{n} \log \Gamma(\alpha_{c,i}) + \sum_{i=1}^{n} (\alpha_{c,i} - 1) \log w_{i,c} \right]$$

$$+ \log \sum_{y_c \in \mathcal{Y}} \exp \left( \sum_{i=1}^{n} \left( w_{i,c} \boldsymbol{z}_{i,c}^{\mathrm{T}} \right)^{\top} \cdot \boldsymbol{z}^{\mathrm{I}} \right) \cdot \frac{\mathbb{1}_{\hat{y}_{ji} = y_c}}{\sum_{j'} \mathbb{1}_{\hat{y}_{j'i} = y_c}}$$

However, since Dirichlet priors would introduce additional steps (e.g., estimating concentration parameters $\alpha$), in our preliminary investigation, we used uniform prior to keep simplicity. Despite this simplest setup, our CARPRT prompt reweighting strategy effectively facilitated TTA methods. We leave more systematic explorations of alternative priors (e.g., Dirichlet) into future work.

## H  DETAILED RESULTS FOR HYPERPARAMETER ANAYLSIS

In this section, we present the results of our hyperparameter analysis across all fine-grained datasets. Table 6 shows the accuracy for varying temperature setting. In zero-shot classification, where only test data is available, conventional hyperparameter selection is inherently difficult due to the lack of training or validation data. Following Shu et al. (2018), we aim to identify hyperparameters that exhibit robust and consistent performance across diverse datasets..

As shown in Table 6, a temperature of 3.0 consistently provides strong results across datasets. While it may not be optimal for each dataset, it offers a practical and generalizable choice under the constraints of the zero-shot setting.

Table 6: Accuracy(%) results for varying temperature settings across fine-grained datasets using CLIP-ViT-B/16 backbone. Bold value represents the highest accuracy in each column.

| Temperature | Caltech101 | DTD | EuroSAT | Aircraft | Food101 | Flower102 | Pets | Cars | SUN397 | UCF101 | Average |
|---|---|---|---|---|---|---|---|---|---|---|---|
| 1 | 87.97 | 46.16 | 54.07 | 21.13 | 84.42 | 57.97 | 74.05 | 58.10 | 58.29 | 61.07 | 60.32 |
| 2 | 91.91 | 47.56 | 56.04 | 22.62 | 85.87 | 68.88 | **82.76** | 64.35 | 64.85 | 67.91 | 65.28 |
| 3 | 92.60 | **47.74** | **55.85** | **22.64** | 85.78 | 68.58 | 82.48 | 65.02 | **65.49** | **68.61** | **65.48** |
| 4 | 92.56 | 47.10 | 54.62 | 22.56 | 85.68 | 66.42 | 81.86 | 65.26 | 65.48 | 68.69 | 65.02 |
| 5 | **92.72** | 47.08 | 54.03 | 22.62 | **85.62** | 65.68 | 81.74 | **65.38** | 65.39 | 68.40 | 64.87 |
| 10 | 92.68 | 47.24 | 52.59 | 22.30 | 85.47 | 65.05 | 81.34 | 63.23 | 63.23 | 67.53 | 63.86 |

Table 7: Accuracy (%) comparison between our method and baselines on CIFAR-10 using the CLIP-ViT-B/16 backbone. **Bold** values represent the highest accuracy in each column.

| | Balanced Datasets | $\beta = 10$ | $\beta = 50$ | $\beta = 100$ |
|---|---|---|---|---|
| Average | 89.56 | 89.58 | 89.57 | 89.56 |
| ZPE | 89.55 | 90.02 | 90.78 | 91.07 |
| CARPRT | **90.82** | **91.07** | **91.36** | **91.70** |
| Gain from ZPE | +1.27 | +1.05 | +0.58 | +0.63 |
| Gain from Average | +1.26 | +1.49 | +1.79 | +2.14 |

## I EXPERIMENTS ON IMBALANCED DATASETS

In this section, we evaluate the performance of CARPRT on datasets with class imbalances. Following Cao et al. (2019), we manually construct an imbalanced CIFAR-10 (Krizhevsky et al., 2009) dataset using an exponential decay strategy to create various degrees of class imbalance. We use an imbalance factor $\beta$ to describe the severity of the long-tailed distribution, defined as the ratio between the number of training samples in the most frequent class and the least frequent class. Specifically, $\beta$ is given by:

$$\beta = \frac{N_{\max}}{N_{\min}},$$

where $N_{\max}$ and $N_{\min}$ represent the number of training samples in the most frequent and least frequent classes, respectively. We conduct experiments with different imbalance ratios, setting $\beta = 10$, $\beta = 50$, and $\beta = 100$, using the CLIP-ViT-B/16 backbone.

The results shown in Table 7 demonstrate that CARPRT significantly outperforms the average baseline for all degrees of class imbalance. Specifically, CARPRT provides a consistent improvement in performance over ZPE, though the gain decreases as the imbalance factor $\beta$ increases. This decreasing gain may be attributed to the global nature of the ZPE weight estimation, which remains effective even under a higher imbalance. ZPE calculates a single weight for the entire dataset, capturing the overall distribution and maintaining reasonable performance, even when certain classes are underrepresented.

In contrast, CARPRT uses a per-class weighting strategy, which allows better adaptation to individual class characteristics, which is highly effective in balanced or moderately imbalanced settings. However, when the class imbalance becomes severe, the challenge arises for classes with very few samples (e.g., only 10 samples). In these cases, the reliability of CARPRT's weight estimates decreases as a result of insufficient data, impacting performance.

## J IMPACT OF TEMPLATE QUALITY

In this section, we investigate the impact of template quality on ImageNet classification tasks. Specifically, we explore how different prompt template pools influence performance by evaluating two newly generated template pools alongside the original templates on the ImageNet datasets. Specifically, Pool1 was generated using Claude 3.5 (Anthropic, 2024) to produce 300 templates tailored to the ImageNet label space. Each category in Pool1 consists of 100 prompt templates structured in descriptive formats, such as *"A photo of a ", "A photo of a ", "The type of "*. These templates aim to incorporate task-specific context and improve the alignment between the prompts and ImageNet categories. Pool2, on the other hand, was constructed using Phi 3.1 (OpenAI, 2024)

to create highly descriptive templates. For each ImageNet category, Phi 3.1 generated five detailed prompts, resulting in a total of 5,000 templates across all categories. These templates focus on providing class-specific descriptive information, enabling a more precise and nuanced interaction with the underlying vision-language model. These additional template pools were evaluated on ImageNet dataset compared to the original templates (Pool0), as shown in Table 8.

Table 8: Accuracy (%) comparison across different template pools using ZPE and CARPRT methods on ImageNet classification.

| Pool | Method | ImageNet Acc. (%) | Perf. Comparison |
|------|--------|-------------------|------------------|
| Pool0 | ZPE | 67.42 | – |
| | CARPRT | **67.81** | +0.39 |
| Pool1 | ZPE | 68.18 | – |
| | CARPRT | **68.36** | +0.18 |
| Pool2 | ZPE | 68.14 | – |
| | CARPRT | **68.79** | +0.65 |

For Pool1, This pool targets more task-specific information by generating templates with respect to the ImageNet label space. This leads to performance improvements for both ZPE and CARPRT prompt reweighting strategies compared to Pool0. On the other hand, the generated templates in Pool2 incorporate more class-specific descriptive information. CARPRT benefits significantly from these templates, achieving greater performance gains compared to ZPE. This highlights the effectiveness of class-aware prompt reweighting in leveraging descriptive templates.

## K  COMBINING CLASS-AWARE PROMPT REWEIGHTING WITH PROMPT TUNING METHOD.

*Prompt tuning* has recently become a powerful technique for adapting CLIP and other pre-trained vision-language models to downstream tasks. By learning optimal prompts that guide the model's understanding of new data, prompt tuning has shown remarkable effectiveness (Zhou et al., 2022b;a; Khattak et al., 2023b). ProDA optimizes prompt distributions to improve few-shot performance by training a set of learnable invisible prompt embeddings. While CARPRT is primarily designed to reweight visible prompt templates, our approach is not restricted to visible prompts. In this section, we also apply class-aware reweighting to the invisible prompts trained by ProDA, making our method capable of enhancing performance in various prompt tuning scenarios.

Our CARPRT method could enhance the ProDA framework by introducing a class-aware reweighting technique that adjusts the influence of each prompt based on the underlying class structure. Specifically, before each iteration of ProDA's prompt distribution learning, we use CARPRT to update the weights, which then guide the model's logit outputs for training the prompts. As the problem setting transitions from zero-shot to few-shot, our approach adapts by refining the weight estimation. Specifically, we use ground truth labels instead of the pseudo labels for weight estimation, as shown in the following replacement for Eq. (15):

$$w'_{i,c} = \frac{\sum_{j=1}^{m} s_{j,i,c} \mathbb{1}_{y_j=y_c}}{\sum_{j=1}^{m} \mathbb{1}_{y_j=y_c}},$$

(18)

where $y_j$ is the ground truth label of the sample $j$. The results show in Table 9 demonstrate that our method provides notable improvements in most data sets, highlighting the effectiveness of our class-aware prompt reweighting mechanism.

## L  ETHIC STATEMENT

This research adheres to the ICLR Code of Ethics, ensuring that all practices comply with ethical standards regarding data privacy, research integrity, and fairness. The study does not involve human

Table 9: Accuracy (%) comparison between our method and the baseline on fined-grained datasets using the CLIP-ViT-B/16 backbone. **Bold** values represent the highest accuracy in each raw.

|  | ProDA | ProDA + CARPRT |
|---|---|---|
| Caltech101 | 91.3 | **95.4** |
| DTD | **70.1** | 69.6 |
| EuroSAT | **84.3** | 83.4 |
| Aircraft | 36.6 | **36.9** |
| Food101 | 82.4 | **88.1** |
| Flower102 | 95.5 | **95.6** |
| Pets | 90.0 | **93.7** |
| Cars | 75.5 | **78.6** |
| Average | 78.2 | **80.2** |

or animal subjects, and all datasets mentioned in Appendix B utilized in the experiments are publicly available and anonymized, eliminating any potential privacy concerns. We have taken careful steps to avoid any potential bias or unfairness in our methodologies and experimental procedures. The algorithms and models developed in this work are designed to be transparent and reproducible, and any ethical concerns related to the broader applications of the research have been addressed appropriately.

