# OpenReview forum: "CARPRT: Class-Aware Prompt Reweighting for Pre-Trained Vision-Language Models"
_ICLR.cc/2025/Conference — Submitted to ICLR 2025_

### Official Review · Reviewer_CBgL · 2024-10-23

**Soundness:** 3
**Presentation:** 3
**Contribution:** 3
**Rating:** 6
**Confidence:** 4

**Summary:**

This paper studies class-aware prompt reweighting strategy for vision-langauge models. The idea is that when doing prompt ensembling, using the same weighting vector for all classes overlooks the diverse characteristics of different classes, and may yield suboptimal results. Further, the authors present a probabilistic viewpoint based on Bayes' Theorem to show the conditional independence assumption between the class and weights in the class-shared weighting method. CARPRT is proposed that obtains weight vectors based on the relevance score to prioritize the most relevant prompts for each class. The efficacy of CARPRT is verified using CLIP model over multiple benchmarks.

**Strengths:**

- This paper studies the important prompt ensembling task. With the popularity of zero-shot image classification with multimodal models such as CLIP, the proposed method/idea can be useful to a wide audience.

- The idea of class-aware prompt weighting is simple, but the authors present a probabilistic viewpoint based on Bayes' theorem to show its advantage over class-independent prompt weighting.

- The final CARPRT method is a simple implementation that adheres to the key principles established in the probabilistic analyses. Over 10 fine-grained datasets and 3 ImageNet variants, it shows an improvement in accuracy over ZPE. Both ViT and ResNet backbones are considered. Hyper-parameter studies are adequate.

- The paper is well-written and easy to follow.

**Weaknesses:**

- The proposed CARPRT adheres to the key principles established in Section 3, but the detailed correspondences are not explicitly presented. If I understand it correctly, the marginalization in Eq.(6) over $W$ reduces to a point estimation of $W$ for the following analysis. $Pr(W|P)$ is not reflected in the implementation. $Pr(y_c|W,P)$ is approximated by Eq.(9). The final weights $w$ in Eq.(16) seem to be the posterior approximation of Eq.(7), but how to derive these formulations (from Eq.(13) to Eq.(16)) are omitted in the paper. The authors may want to add a section of mathematical derivations in the appendix that can assist audiences to understand how CARPRT is designed from the probabilistic forumations.

- Using class-aware prompt weighting theoretically is better than class-independant prompt weighting, but its robustness is also a practical concern. The authors also mention the influences of pseudo-labels and prompt template pool, but relevant empirical analyses are not provided. For example, is CARPRT more sensitive to noisy pseudo-labels than ZPE? (perhaps we can manually inject noises into pseudo-labels to understand the influences) What happens when fine-grained datasets have more or less diverse template pool? Is it really necessary to combine templates from different datasets for a particular test dataset?

**Questions:**

- Please clarify the correspondances between CARPRT formulations and probabilistic derivation.

- In Appendix D, the authors discuss several forms of $Pr(W|P)$. Could the authors explain how these priors can be obtained and used, for example, in the Test-Time Adaptation situation mentioned in the paper?

- I am confused about the toy example in Proposition 3. Why all the three (Ln 880-883) have $y_1$ as their variates? How Ln 886-889 are derived?

- In Ln 431, a typo in 'iaccuracy'.

---

> ### Author Response · Authors · 2024-11-21
> **Rebuttal by Authors**
>
> **Response to correspondence between CARPRT formulation and the probabilistic framework (W1 & Q1):**
>
> Thanks for raising this up.
> We agree with that we should have been set up clearer correspondence between CARPRT formulation and the established probabilistic framework in Section 3.
>
> Concretely, the practical implementation Eqs.(13-16) align with Eqs.(7-11) in the following manner.
>
> **Score Calculation**. (Eq. (13)) implements the likelihood term $\\Pr(\\boldsymbol{x}\_j|y_c, W, \\mathbb{P})$ from Eq. (11) by defining $s\_{j,i,c} = \frac{\exp(a_{j,i,c} / \lambda)}{ \sum_{y \in \mathcal{Y}} \exp(a_{j,i,c} / \lambda)}$.
> This formulation align with the EBM in Eq. (1) by
> - using cosine similarity $a_{j, i, c}$ as the negative energy term and
> - normalizing through softmax to obtain proper probabilities
>
> **Weight Calculation**. (Eq. (15-16)) correspond to estimating $\\Pr(W | \\mathbb{P}, \\mathbb{D})$ from Eq.(8) through a two-step process.
> - Eq. (15) first obtains the pseudo-labels for samples as the empirical estimates $\hat{\Pr}(y\_c | W, \mathbb{P})$ (i.e., Eq. (9)). It then estimates intermediate weights by aggregating scores across pseudo-labeled samples by multiplying the scores $\Pr(\boldsymbol{x}_j|y_c, W, \mathbb{P})$ (i.e., $s\_{j,i,c}$) with $\hat{\Pr}(y_c | W, \mathbb{P})$
> - Eq. (16) applies softmax to ensure the resulting weights form a valid probability distribution over prompts for each class, which satisfies the simplex constraint implied by our probabilistic framework.
>
> We appreciate your valuable comments and will merge this into the revision.

---

> > ### Comment · Reviewer_CBgL · 2024-11-25
> >
> > I appreciate the response. I have a quick question: Eq.(13) is likelihood over feature $x_j$, and $s_{j,i,c}$ is more like a posterior probability over label $y_c$. There seems to be a gap between Eq.(13) and $s_{j,i,c}$.

---

> > > ### Author Response · Authors · 2024-11-25
> > > **Response to follow-up questions**
> > >
> > > > There seems to be a gap between the likelihood over feature $\boldsymbol{x}\_j$ and $s\_{j, i, c}$.
> > >
> > > Thanks very much for raising this follow-up question. We recognize the concern regarding the gap between the likelihood of $\boldsymbol{x}_j$ (Eq. (11)) and originally presented $s\_{j, i, c}$.
> > > Upon revisiting the text, we identified a **presentation error** in the original submission.
> > >
> > > 1. Clarification of the issue:
> > > Eq. (13), as initially written, incorrectly presents $s\_{j, i, c}$.
> > > In fact, the value of interest here should have been $a\_{j, i, c}$, i.e., the cosine similarity between $\boldsymbol{z}\_{j}^{\rm I}$ and $\boldsymbol{z}\_{i, c}^{\rm T}$.
> > > The $a\_{j, i, c}$ represents the unnormalized likelihood of $\boldsymbol{x}_{j}$, and **no softmax operation should be applied to it**.
> > >
> > > 2. Correction:
> > > The corrected Eq. (13) should yield $a\_{j, i, c}$, not $s\_{j, i, c}$. With this adjustment, there is **no gap** between Eq. (11) (in the theoretical framework) and Eq. (13) (in the practical implementation).
> > >
> > > 3. Consistency with implementation:
> > > Our implementation (lines 105-120 in `utils.py` of the submitted code) reflects this corrected understanding -- there is no softmax applied after getting the logits of CLIP.
> > >
> > > We sincerely apologize for the oversight and the associated confusion it caused.
> > > The revision will address this error explicitly to ensure clarity and consistency very soon.
> > >
> > > ---
> > >
> > > > injecting noises into pseudo-labels by randomly flipping a portion of pseudo-labels.
> > >
> > > We appreciate your recognition of our effort in addressing this question, and thank you very much for proposing this interesting idea! This is a valuable suggestion for assessing the robustness of prompt ensembling methods.
> > >
> > > In the CARPRT framework, pseudo-labels are generated based on CLIP's prediction across image and multiple text embeddings.
> > > Adding controlled noise could, intuitively, serve as a mechanism to test and potentially improve robustness against prompt sensitivity.
> > > However, as we noted in our previous response, accurately defining pseudo-label accuracy within this setting presents challenges.
> > > We plan to leave a more systematic exploration of noisy pseudo-labels to future works and will include a discussion of this idea in our paper.
> > >
> > > Once again, We sincerely thank you for your careful reading and insightful suggestions, which have significantly helped us to refine our work.

---

> ### Author Response · Authors · 2024-11-21
> **Rebuttal by Authors**
>
> **Response to influences of pseudo-labels and prompt template pool (W2):**
>
> Thank you very much for the question and suggestion.
>
> **The influences of pseudo-labels.**  Upon further investigation, we realized that defining pseudo-label accuracy is non-trivial in our problem setting compared to standard classification tasks. As detailed in Section 4.2, we create a pseudo-label set $\hat{\mathbb{Y}} = \\{ \hat{y}_{j, i} \\}^{m, n}\_{j=1, i=1}$ for each sample $\boldsymbol{x}_j$, where $\hat{y}_c$ corresponds to the class with the highest relevance score for each of $n$ templates.
> This could lead to **multiple pseudo-labels being predicted to the same sample by different templates**, challenging the clear definition of conventional pseudo-label accuracy.
>
> As an alternative metric, we "define" a pseudo-label accuracy as the proportion of templates among all templates that correctly predict the ground truth label for each sample. Using this definition, we compute the Pearson correlation coefficient between pseudo-label accuracy and the test accuracy improvement of CARPRT over the Equal Weight baseline across all fine-grained datasets. The resulting correlation coefficient of 0.5021 suggests a weak to moderate positive relationship.
>
> This finding indicates that, while pseudo-label accuracy may contribute to the effectiveness of CARPRT, the relatively weak correlation with test accuracy improvement suggests it may not be the dominant factor. We appreciate this observation and acknowledge the importance of investigating additional factors influencing our method's success.
>
> Regarding your suggestion of "injecting noises into pseudo-labels", we are not that familiar with this strategy and thus have no concrete idea of how to implement this within our setting. Could you please provide more details or references regarding these methods so we can understand your suggestion better?
>
> ---
> Alternatively, inspired by your suggestion, we conduct additional experiments to **investigate whether and how the quality of the template pool matters**.
> **The influences of the template pool**.
> We performed additional experiments to further investigate the impact of template quality more comprehensively.
> We attempted to generate **two new** template pools (Pool1 and Pool2) using LLMs.
> The details and results are summarized below:
> - **Pool0**: Original templates (as reported in the paper).
> - **Pool1**: We used [Claude 3.5](https://www.anthropic.com/news/claude-3-5-sonnet) to generate 300 templates by introducing the ImageNet’s label space as context, consisting of 100 templates each in the following formats: *"A photo of a {}, a type of XXX."*, *"A XXX photo of a {}."* and *""A XXX of a {}."*
> - **Pool2**: Using [Phi 3.1](https://huggingface.co/bartowski/Phi-3.1-mini-128k-instruct-GGUF), we generated five descriptive templates per ImageNet category, resulting in a total of 5,000 templates.
>
> | Pool   | Method | ImageNet Acc. | Perf. comparison |
> |--------|--------|----------------|------|
> | Pool0  | ZPE    | 67.42          |      |
> |        | CARPRT | 67.81          | +0.39 |
> | Pool1  | ZPE    | 68.18          |      |
> |        | CARPRT | 68.36          | +0.18 |
> | Pool2  | ZPE    | 68.14          |      |
> |        | CARPRT | 68.79          | +0.65 |
>
> **Key observations**.
> 1. Template Quality Matters:
>     - Pool1: Task-specific templates aligned with ImageNet labels improved both ZPE and CARPRT performance.
>     - Pool2: Class-specific templates resulted in the most significant gains for CARPRT, demonstrating its ability to utilize detailed, class-aware information more effectively than ZPE, highlighting the necessity of class-aware prompt reweighting.
>
> 2. Consistent CARPRT Gains:
>     - CARPRT consistently outperformed ZPE across all template pools, particularly with high-quality, class-specific templates.
>
> This indicates that enhancing template quality indeed yields better performances, which could be one of the promising research directions in the future.
>
> We hope this additional finding could help to address your concern.

---

> > ### Comment · Reviewer_CBgL · 2024-11-25
> >
> > Regarding your suggestion of "injecting noises into pseudo-labels" --> My idea was to randomly flip a portion of pseudo-labels (say 10%) into other values during training process. Thus we can see how CARPRT is robust to wrong pseudo-labels.
> >
> > Nevertheless, I appreciate the authors' response on this question.

---

> ### Author Response · Authors · 2024-11-21
> **Rebuttal by Authors**
>
> > *Due to the formatting inconsistency between LaTeX and Markdown syntax within the OpenReview system, we find it difficult to include all the math formulas in this response directly. To this end, we have incorporated detailed explanations and derivations **in the revision**, please kindly refer to the updated PDF for the full information. We appreciate your understanding.*
>
> **Response to how different priors can be obtained and used (Q2):**
>
> Thanks for the question! When prompt weights can be updated continuously, such as in TTA settings, different priors (e.g., uniform, global Dirichlet, or class-specific Dirichlet) define initial beliefs about weight distributions before observing test data. In **Appendix G**, we supplement a comprehensive discussion on the derivation and impact of various priors (e.g., uniform, global Dirichlet, class-specific Dirichlet) on weight updates in our proposed method.
> However, since Dirichlet priors would introduce additional steps (e.g., estimating concentration parameters $\alpha$), in our preliminary investigation, we used a uniform prior to keep simplicity.
> Despite this most simplest setup, our CARPRT prompt reweighting strategy effectively facilitated TTA methods.
> We leave a more systematic exploration of alternative priors (e.g., Dirichlet) into future work.
>
> **Response to corrections and typo (Q3 & Q4):**
>
> Thank you very much for pointing them out! Ln 880-883 are indeed typos, which should have been $y_1, y_2, y_3$ for each line. We have also completely revised the proof in **Appendix E**. Thank you once again for helping us to improve the quality of our manuscript.

---

> ### Author Response · Authors · 2024-11-24
> **Reminder - Discussion Stage Closing Soon - 24 November**
>
> Dear Reviewer CBgL,
>
> We appreciate the time and effort that you have dedicated to reviewing our manuscript.
>
> We have carefully addressed all your queries. Could you kindly spare a moment to review our responses?
>
> Have our responses addressed your major concerns?
>
> If there is anything unclear, we will address it further. We look forward to your feedback.
>
> Best regards,
>
> Authors of Submission6458

---

> ### Author Response · Authors · 2024-11-30
> **Looking forward to your responses or further suggestions/comments!**
>
> Dear Reviewer CBgL,
>
> Following your valuable comments and suggestions, we have provided a detailed reply to address additional questions and concerns raised after our initial reply. We are happy to discuss them with you in the openreview system if you feel that there still are some concerns/questions. We also welcome new suggestions/comments from you!
>
> On the other hand, if our reply has adequately addressed the concerns, we would greatly appreciate it if you could consider updating the review score accordingly.
>
> Thank you once again for your time and effort in reviewing our work. We look forward to hearing from you!
>
> Best regards,
> Authors of Submission6458

---

> > ### Comment · Reviewer_CBgL · 2024-12-03
> >
> > Thanks for the followup. I keep my positive score.

---

### Official Review · Reviewer_a8Wx · 2024-11-01

**Soundness:** 3
**Presentation:** 3
**Contribution:** 3
**Rating:** 6
**Confidence:** 4

**Summary:**

This paper proposes a category-aware prompt re-weighting (CARPRT) method to improve the performance of pre-trained visual language models (VLMs) on zero-shot image classification tasks. It first finds the limitations of existing prompt re-weighting methods, and explains the root cause from a probabilistic analysis perspective. The proposed CARPRT calculates the relevance scores of prompt-class pairs and determines the optimal prompt weight vector for each class based on these scores. The experimental results show that CARPRT outperforms existing re-weighting strategies on various image classification benchmarks.

**Strengths:**

1. The proposed CARPRT introduces a class-specific weighting mechanism, significantly improving the alignment of prompts with class-specific information.
2. The paper provides a solid theoretical foundation, addressing the conditional independence limitations in previous methods.
3. By automating the prompt reweighting process, the method reduces dependency on manually crafted prompts.

**Weaknesses:**

1. The approach introduces additional computational complexity while the improvement is extremely limited (less than 1%) on part of situations. Moreover, only the weighted baselines are compared, I doubt that whether it can enhance existing prompt engineer methods.
2. The effectiveness of CARPRT is tied to the quality and diversity of the prompt template pool, is it effective for learned prompts?

**Questions:**

see weaknesses

---

> ### Author Response · Authors · 2024-11-21
> **Rebuttal by Authors**
>
> **Response to Complexity Concern (W1):**
>
> Thank you for raising this concern.
> We would like to first emphasize that **CARPRT introduces no significant computational overhead compared to ZPE**, while achieving consistently better performance on almost all datasets.
> To address your concern, we provide a concise complexity analysis below.
>
> The computational complexity of reweighting methods (including class-wise, prompt-wise, and class-aware prompt reweighting) arises from three main sources: embedding calculations using CLIP, similarity score computations, and weight matrix calculation. We note that the **embedding calculation is the most computationally expensive part**, as each image and all prompt-class combinations must pass through the image and text encoder.
> *This computation is shared by all prompt reweighting methods*.
> We denote the complexity of the text (image) encoder pass as $\Delta_T$ ($\Delta_I$). Thus, the overall complexity of CARPRT is:
>
> $O(m\Delta_I + nC\Delta_T + mnC + nC) \approx O(m\Delta_I + nC\Delta_T),$
>
> and for prompt-wise reweighting:
>
> $O(m\Delta_I + nC\Delta_T + mnC + n) \approx O(m\Delta_I + nC\Delta_T).$
>
> We note that $\Delta$ (encoder complexity) is significantly larger than $m$ due to the multi-layered Transformer architecture, making encoder computations the **dominant cost**. Furthermore, $m$ (the number of images) is much larger than $n$ (the number of prompts) and $C$ (the number of classes), underscoring that the main burden lies in the encoder passes rather than subsequent operations.
>
> In conclusion, CARPRT matches ZPE in computational efficiency while delivering superior results, as verified across diverse datasets. We will include this complexity analysis in the revision to clarify this point.
>
> We hope this clarification addresses your concern.

---

> ### Author Response · Authors · 2024-11-21
> **Rebuttal by Authors**
>
> **Response to integration to Prompt Engineering methods (W1 & W2):**
>
> Thank you for the question. We demonstrate that our method **can enhance existing prompt engineering** methods.
> We extend our prompt reweighting strategy CARPRT to ProDa [1], a *prompt tuning* method that trains multiple invisible text prompts per dataset.
> Since the problem setting transitions from zero-shot to few-shot, our estimation in Eq. (15) evolves from pseudo-labels to ground-truth labels.
> Our combined approach demonstrated enhanced performance across various datasets, as shown in the table below:
> | Dataset      | ProDA  | ProDA + CARPRT |
> |--------------|--------|----------------|
> | Caltech101   | 91.3   | **95.4**       |
> | DTD          | **70.1** | 69.6         |
> | EuroSAT      | **84.3** | 83.4         |
> | Aircraft     | 36.6   | **36.9**       |
> | Food101      | 82.4   | **88.1**       |
> | Flower102    | 95.5   | **95.6**       |
> | Pets         | 90.0   | **93.7**       |
> | Cars         | 75.5   | **78.6**       |
> | **Average**  | 78.2   | **80.2**       |
>
> This indicates that our method can effectively complement and enhance existing prompt engineering techniques.
> We will include these results in the revised manuscript.
>
> ---
>
> In addition to learnable prompts, we have illustrated that our method yields better results once we have obtained a better template pool.
> Please check  [this response](https://openreview.net/forum?id=fRpAUgKJhT&noteId=JLvxUZxKKN) for your reference.
>
> We hope our additional results on both learnable prompts, and higher-quality prompt templates sort out this question.
>
> [1] Lu, Y., Wei, F., Lu, W., and Yu, D. "Prompt Distribution Learning." *CVPR*, 2022.

---

> > ### Comment · Reviewer_a8Wx · 2024-11-24
> > **Thanks for the reply**
> >
> > 1. The complexity analysis is good. Is there more statistical data to support it？
> > 2. ProDa is the learnable prompts, and Where are the  additional results for higher-quality prompt templates ?

---

> > ### Author Response · Authors · 2024-12-04
> > **[Update 04/Dec/2024] Results Correction!**
> >
> > Dear reviewer a8Wx,
> >
> > During our attempt to quickly reproduce the implementation of ProDa and report additional results within a very short period, we encountered some minor issues in our implementation that required adjustments. As a result, we have updated our findings. The corrected results are shown below:
> >
> > | Dataset       | RN50-ep50 ProDA | **RN50-ep50 ProDA+Ours** | VIT16-ep50 ProDA | **VIT16-ep50 ProDA+Ours** |
> > |---------------|---------------------------------|--------------------------|----------------------------------|---------------------------|
> > | Caltech101    | 90.79                           | **91.25**               | 95.15                           | **95.62**                |
> > | DTD           | 69.47                           | **70.05**               | 70.46                           | **71.73**                |
> > | EuroSAT       | 78.36                           | **80.54**               | 75.04                           | **78.52**                |
> > | Aircraft      | 36.54                           | **36.69**               | 35.87                           | **37.92**                |
> > | Food101       | 80.15                           | **80.41**               | 88.12                           | **88.14**                |
> > | Flower102     | 94.47                           | **95.25**               | 95.4                            | **96.43**                |
> > | Pets          | 89.14                           | **89.53**               | 93.89                           | **93.84**                |
> > | Cars          | 74.25                           | **74.37**               | 83.21                           | **83.25**                |
> > | SUN397        | 69.36                           | **70.04**               | 76.21                           | **76.79**                |
> > | UCF101        | 75.71                           | **76.14**               | 83.74                           | **84.54**                |
> > | ImageNet      | 65.06                           | **65.27**               | 71.96                           | **72.07**                |
> > | **Average**   | **74.85**                       | **75.41**               | **79.00**                       | **79.90**                |
> >
> > In the above table:
> > - **RN50-ep50** refers to results using **CLIP-ResNet50**, trained for 50 epochs.
> > - **VIT16-ep50** refers to results using **CLIP-ViT-B/16**, trained for 50 epochs.
> >
> > The corrected results demonstrate that CARPRT consistently outperforms ProDA under a fair setting across all datasets. This consistent improvement highlights CARPRT’s robustness and its effectiveness in complementing ProDA to enhance performance in various scenarios.
> >
> > We sincerely apologize for the oversight and any confusion caused. These updated results will be included in the revised manuscript. We appreciate your understanding and continued support.
> >
> > Best regards,
> >
> > Authors of Submission6458

---

> ### Author Response · Authors · 2024-11-24
> **Further Statistical Evidence for Complexity Analysis and Experiment Results with Higher-Quality Prompt Templates**
>
> Thank you for your comments. We appreciate your positive feedback on the complexity analysis and your interest in additional statistical evidence. Below, we provide further statistical evidence to support complexity analysis and the experiment results for for higher-quality prompt templates .
>
> > The complexity analysis is good. Is there more statistical data to support it?
>
> Thank you for the question. To validate the complexity analysis, we compared the actual computation times for CARPRT and ZPE across datasets. The table below summarizes the results (time in seconds):
>
> | Dataset       | CARPRT Time (s) | ZPE Time (s) |
> |---------------|-----------------|--------------|
> | Caltech101    | 7.11            | 6.55         |
> | DTD           | 7.78            | 7.75         |
> | EuroSAT       | 11.55           | 11.35        |
> | Aircraft      | 18.18           | 18.18        |
> | Food101       | 50.97           | 50.15        |
> | Flowers102    | 9.04            | 9.16         |
> | Oxford Pets   | 9.58            | 9.64         |
> | Cars196       | 22.81           | 22.20        |
> | SUN397        | 107.50          | 104.42       |
> | UCF101        | 7.37            | 7.57         |
>
> For instance, on EuroSAT (\(m=8100, C=10\)), CARPRT took 11.55 s, and ZPE took 11.35 s, showing negligible overhead. Similarly, on SUN397 (\(m=19,850, C=397\)), CARPRT (107.50 s) closely matched ZPE (104.42 s). This consistency across datasets confirms that encoder operations dominate computational costs, as reflected in the complexity formulas.
>
> > ProDa is the learnable prompts, and Where are the additional results for higher-quality prompt templates ?
>
> Thanks for your comments. We also conduct the experiment over the higher-quality prompt templates. We attempted to generate **two new** template pools (Pool1 and Pool2) using LLMs.
> The details and results are summarized below:
> - **Pool0**: Original templates (as reported in the paper).
> - **Pool1**: We used [Claude 3.5](https://www.anthropic.com/news/claude-3-5-sonnet) to generate 300 templates by introducing the ImageNet’s label space as context, consisting of 100 templates each in the following formats: *"A photo of a {}, a type of XXX."*, *"A XXX photo of a {}."* and *""A XXX of a {}."*
> - **Pool2**: Using [Phi 3.1](https://huggingface.co/bartowski/Phi-3.1-mini-128k-instruct-GGUF), we generated five descriptive templates per ImageNet category, resulting in a total of 5,000 templates.
>
> | Pool   | Method | ImageNet Acc. | Perf. comparison |
> |--------|--------|----------------|------|
> | Pool0  | ZPE    | 67.42          |      |
> |        | CARPRT | 67.81          | +0.39 |
> | Pool1  | ZPE    | 68.18          |      |
> |        | CARPRT | 68.36          | +0.18 |
> | Pool2  | ZPE    | 68.14          |      |
> |        | CARPRT | 68.79          | +0.65 |
>
> **Key observations**.
> 1. Template Quality Matters:
>     - Pool1: Task-specific templates aligned with ImageNet labels improved both ZPE and CARPRT performance.
>     - Pool2: Class-specific templates resulted in the most significant gains for CARPRT, demonstrating its ability to utilize detailed, class-aware information more effectively than ZPE, highlighting the necessity of class-aware prompt reweighting.
>
> 2. Consistent CARPRT Gains:
>     - CARPRT consistently outperformed ZPE across all template pools, particularly with high-quality, class-specific templates.
>
> The results indicate that enhancing template quality does yield performance improvements.
>
> We hope these revisions clearly address your concerns.

---

> > ### Comment · Reviewer_a8Wx · 2024-11-25
> > **Thanks for the reply**
> >
> > Thank you for your response and it addresses some of my concerns. I will raise the score.

---

> ### Author Response · Authors · 2024-11-25
> **Glad to hear that your concerns are addressed! Thanks for raising your score!**
>
> Dear Reviewer a8Wx,
>
> Thank you very much for raising the score, and we are glad that your concerns are addressed!
>
> We will continuously merge the suggestions and modifications into our revision. We appreciate your time and efforts in helping us to improve the quality of our paper.
>
> Best,
>
> Authors of submission 6458

---

### Official Review · Reviewer_6Qg1 · 2024-11-04

**Soundness:** 3
**Presentation:** 2
**Contribution:** 3
**Rating:** 6
**Confidence:** 3

**Summary:**

The paper presents CARPRT, a method to automatically choose and assign weights to a given set of standard prompts when used for zero shot image classification. CARPRT specifically addresses the limitation of the current zero-shot method that uses a fixed weighting vector for the given set of prompts for all classes. Instead, it adjusts prompt weights for each class based on image-prompt relevance by leveraging pseudo-labeling. Experiments on multiple benchmarks show that CARPRT improves zero-shot classification accuracy, especially in fine-grained classification tasks.

**Strengths:**

**Well motivated**: CARPRT effectively demonstrates the need for class-aware weights, showing that a shared weight for prompts for all classescan hinder zero-shot VLM performance.

**Simple, mathematically sound probabilistic framework**: The paper uses an energy-based probabilistic framework to model underlying class distributions for prompt reweighting, enhancing the interpretability of CARPRT’s approach.

**Improvements on Fine-grained datasets**: The method achieves state-of-the-art results on multiple datasets, notably improving accuracy on some fine-grained datasets like EuroSAT and Flowers102.

**Weaknesses:**

**W1 Limited Improvement on ImageNet Variants**: The performance gains on datasets like ImageNet-A and ImageNet-R are modest, suggesting that CARPRT’s advantages may be less pronounced on general image datasets compared to fine-grained datasets. more testing on other large datasets (e.g. Places 365) would help better evaluate the methods sginficance for large datasets

**W2 - Pseudo-Label Accuracy Dependency**: The method depends on accurate pseudo-labels to derive class-specific weights, which can be problematic in scenarios where pseudo-labeling quality is low. A more detailed study on how this affects performance might help increase the utility of the method.

**Questions:**

1. Did you try using models other than Energy based ones to model the underlying class distributions (e.g. GMMs, Local Manifold models) ? It would be interesting to see how/ if the choice makes a difference.
2. How well does CARPRT work on fine-grained datasets that have class imbalances ? Does using a different weights for each class improve performance here too ?

---

> ### Author Response · Authors · 2024-11-21
> **Rebuttal by Authors**
>
> **Response to Limited Performance Gains (W1):**
>
> Thank you for your comments. In our paper (Section 5.2 and 6), we have noted that this limited improvement may be associated with the quality of the prompt templates, as well as the intrinsic difficulty and task complexity of ImageNet classification tasks.
>
> In response, we performed additional experiments to further investigate the impact of template quality more comprehensively.
> We attempted to generate **two new** template pools (Pool1 and Pool2) using LLMs.
> The details and results are summarized below:
> - **Pool0**: Original templates (as reported in the paper).
> - **Pool1**: We used [Claude 3.5](https://www.anthropic.com/news/claude-3-5-sonnet) to generate 300 templates by introducing the ImageNet’s label space as context, consisting of 100 templates each in the following formats: *"A photo of a {}, a type of XXX."*, *"A XXX photo of a {}."* and *""A XXX of a {}."*
> - **Pool2**: Using [Phi 3.1](https://huggingface.co/bartowski/Phi-3.1-mini-128k-instruct-GGUF), we generated five descriptive templates per ImageNet category, resulting in a total of 5,000 templates.
>
> | Pool   | Method | ImageNet Acc. | Perf. comparison |
> |--------|--------|----------------|------|
> | Pool0  | ZPE    | 67.42          |      |
> |        | CARPRT | 67.81          | +0.39 |
> | Pool1  | ZPE    | 68.18          |      |
> |        | CARPRT | 68.36          | +0.18 |
> | Pool2  | ZPE    | 68.14          |      |
> |        | CARPRT | 68.79          | +0.65 |
>
> **Key observations**.
> 1. Template Quality Matters:
>     - Pool1: Task-specific templates aligned with ImageNet labels improved both ZPE and CARPRT performance.
>     - Pool2: Class-specific templates resulted in the most significant gains for CARPRT, demonstrating its ability to utilize detailed, class-aware information more effectively than ZPE, highlighting the necessity of class-aware prompt reweighting.
>
> 2. Consistent CARPRT Gains:
>     - CARPRT consistently outperformed ZPE across all template pools, particularly with high-quality, class-specific templates.
>
> The results indicate that enhancing template quality does yield performance improvements.
>
> **Plans for Places365 Evaluation**.
> We acknowledge the importance of testing CARPRT on a diverse, large-scale dataset like Places365.
> However, preparing (download and setup) the dataset is currently underway (it takes more than 1 week), and we will report these results in the revision once we obtained them.

---

> ### Author Response · Authors · 2024-11-21
> **Rebuttal by Authors**
>
> **Response to Pseudo-Label Accuracy Dependency (W2):**
>
> We appreciate your insightful observation.
> Upon further investigation, we realized that defining pseudo-label accuracy is non-trivial in our problem setting compared to standard classification tasks.
> As detailed in Section 4.2, we create a pseudo-label set $\hat{\mathbb{Y}} = \\{ \hat{y}_{j, i} \\}^{m, n}\_{j=1, i=1}$ for each sample $\boldsymbol{x}_j$, where $\hat{y}_c$ corresponds to the class with the highest relevance score for each of $n$ templates.
> This could lead to **multiple pseudo-labels being predicted to the same sample by different templates**, challenging the clear definition of conventional pseudo-label accuracy.
>
> As an alternative metric, we "define" a pseudo-label accuracy as the proportion of templates among all templates that correctly predict the ground truth label for each sample. Using this definition, we compute the Pearson correlation coefficient between pseudo-label accuracy and the test accuracy improvement of CARPRT over the Equal Weight baseline across all fine-grained datasets. The resulting correlation coefficient of 0.5021 suggests a weak to moderate positive relationship.
>
> This finding indicates that, while pseudo-label accuracy may contribute to the effectiveness of CARPRT, the relatively weak correlation with test accuracy improvement suggests it may not be the dominant factor. We appreciate this observation and acknowledge the importance of investigating additional factors influencing our method's success.
>
> **An iterative refinement of pseudo-label prediction and weight estimation**.
> Importantly, we find that we can leverage the predicted prompt weights to update pseudo-labels iteratively. Here, pseudo-labels are defined differently from those in the main text. Specifically, for each sample, we compute a single pseudo-label by aggregating predictions across all prompts using the predicted prompt weights. This enables dynamic refinement of both the pseudo-labels and the class-specific weights.
>
> The table below reports the performance comparison of introducing iterative refinement to CARPRT (denoted by ICARPRT).
>
> | Method | Caltech101 | DTD   | EuroSAT | Aircraft | Food101 | Flower102 | Pets  | Cars  | SUN397 | UCF101 | Average |
> |--------|------------|-------|---------|----------|---------|-----------|-------|-------|--------|--------|---------|
> | CARPRT | 92.60      | 47.74 | **55.85** | 22.64   | 85.78   | 68.58     | 82.48 | 65.11 | 65.49  | 68.48  | 65.48   |
> | ICARPRT  | **94.07**  | **48.21** | 53.79 | **23.71** | **87.24** | **70.01** | **86.64** | **65.63** | **67.28** | **70.41** | **66.70** |
>
> We observe that this iterative strategy greatly enhances overall performance across almost all datasets. We will include these updated results in the revision.

---

> ### Author Response · Authors · 2024-11-21
> **Rebuttal by Authors**
>
> **Response to Distribution Modeling (Q1):**
>
> Thank you for your suggestion. We defined Energy-Based Models (EBMs) in Lemma 1 because CLIP's logits naturally align with the EBM likelihood formulation.
> To our knowledge, alternative distribution modeling like Gaussian Mixture Models (GMMs) might not yield better results, as EBMs are versatile and flexible enough (because EBM does not force the likelihood to be normalized, so it is much less restrictive in functional form [1]) to capture a wide range of distributions, including those representable by GMMs. In other words, EBMs can be seen as the generalization form of GMMs.
> Furthermore, we are not that familiar with the Local Manifold models you mentioned. Could you please provide more details or references regarding these methods so we can understand your suggestion better?
>
> **Response to Imbalance Datasets (Q2):**
>
> Thank you for your thoughtful question.
> To evaluate CARPRT under class imbalance, we manually construct an imbalanced CIFAR-10 dataset following previous work [2], using an imbalance ratio $\beta$ that quantifies the degree of imbalance, defined as $\beta = N_{\max} / N_{\min}$,  where $N_{\max}$ and $N_{\min}$ are the number of samples in the majority and minority classes, respectively.  We experiment with different levels of class imbalance by setting $\beta = 10$, $\beta = 50$, and $\beta = 100$.
>
> **Table: Performance Comparison on CIFAR-10 under Different Imbalance Factors**
>
> | Metric               | Balanced Datasets | $\boldsymbol{\beta = 10}$  | $\boldsymbol{\beta = 50}$  | $\boldsymbol{\beta = 100}$  |
> |----------------------|--------------------|---------------------------|---------------------------|----------------------------|
> | **Average**          | 89.56              | 89.58                     | 89.57                     | 89.56                     |
> | **ZPE**              | 89.55              | 90.02                     | 90.78                     | 91.07                 |
> | **CARPRT**           | **90.82**          | **91.07**                 | **91.36**                 | **91.70**                 |
> | **Gain from ZPE**    | +1.27              | +1.05                     | +0.58                     | +0.63                     |
> | **Gain from Average**| +1.26              | +1.49                     | +1.79                     | +2.14                     |
>
> We observe that CARPRT consistently yields the best performance in comparison with the Average baseline and ZPE, across all imbalance levels.
> Compared to the Average baseline, the gains are more evident as the imbalance increases, highlighting CARPRT’s adaptability to such conditions.
> We also notice that the performance gap between CARPRT and ZPE narrows under severe imbalance.
> This gain reduction likely arises from ZPE's reliance on a single global weight for the entire dataset, which seems to remain effective even when some classes contain very few samples (e.g., 10 instances).
> In contrast, CARPRT's class-specific weight estimation could be less reliable for extremely underrepresented classes.
>
> These results highlight CARPRT's robustness and adaptability in imbalanced settings.
> We will include these findings in the revised manuscript to address this important aspect.
>
>
> [1] Song, Y., and Kingma, D. "How to Train Your Energy-Based Models." In *ArXiV*, 2021
>
> [2] Cao, K., Wei, C., Gaidon, A., Arechiga, N., & Ma, T. "Learning imbalanced datasets with label-distribution-aware margin loss." *NeurIPS*, 2019.

---

> ### Author Response · Authors · 2024-11-24
> **Reminder - Discussion Stage Closing Soon - 24 November**
>
> Dear Reviewer 6Qg1,
>
> We appreciate the time and effort that you have dedicated to reviewing our manuscript.
>
> We have carefully addressed all your queries. Could you kindly spare a moment to review our responses?
>
> Have our responses addressed your major concerns?
>
> If there is anything unclear, we will address it further. We look forward to your feedback.
>
> Best regards,
>
> Authors of  Submission6458

---

> > ### Comment · Reviewer_6Qg1 · 2024-11-25
> > **Thanks for the Reply**
> >
> > Thank you for the detailed rebuttal. I think the experiments with higher quality prompts are pretty good and better demonstrate the effectiveness of the method. The experiments with pseudo-label accuracy are interesting and help shed more light on the core of the method.
> > Q1: I agree that EBMs are likely a better model for CLIP's logits.  Including this explanation would help make the paper more clear.
> >
> >
> > Overall, the response addresses most of my concerns and after going through all the other reviews, I elect to keep my accept score.

---

> > > ### Author Response · Authors · 2024-11-25
> > > **Thanks for your accept score and supporting our paper!**
> > >
> > > Dear Reviewer 6Qg1,
> > >
> > > Many thanks for your reply. It is glad to hear that your concerns have been addressed, and also thank you for maintaining your accept score to support us.
> > >
> > > Since the current score is still "borderline", may we know if you mean **you are more likely to vote "weak accept" for us** or still stick on borderline? If the former, we are very happy you recognize our contributions to the field, if the latter, may we know if you have further concerns about our paper? There is still time to discuss, we are delighted to answer any questions or address any concerns from you.
> > >
> > > We are looking forward to your reply!
> > >
> > >
> > > Best regards,
> > >
> > > Authors of Submission 6458

---

### Official Review · Reviewer_rVkh · 2024-11-05

**Soundness:** 2
**Presentation:** 3
**Contribution:** 2
**Rating:** 5
**Confidence:** 4

**Summary:**

This paper studies how to improve the prompting for the CLIP model. The motivation lies in the fact that existing prompting methods ignore the dependence of prompt weights on different classes. The authors first conduct experiments to demonstrate the positive influence of class-specific weights on prediction performance. Then, a class-aware prompt reweighting method CARPRT is proposed. CARPRT calculates the weight matrix based on the relevance scores between images and prompt-class pairs. Experimental results show that CARPRT surpasses ZPE in most cases.

**Strengths:**

- The motivation is intuitive while reasonable. As far as I know, it is the first work applying the class-aware weighted prompt ensemble.
- The paper is well-written and easy to follow. The authors revisit the existing works, present detailed analyses and explanations, and give clear formulations and an algorithm procedure for the proposed method.
- Extensive experimental results are reported.

**Weaknesses:**

- One concern is regarding the computational complexity. Unlike class-wise or prompt-wise reweighting, the proposed method calculates the weight matrix based on two dimensions. Is there any complexity analysis?
- Another concern is regarding the performance gains. It seems that the results in Table 2 showcase marginal improvement compared to ZPE. The performance gains of some datasets in Table 1 are also limited. This may raise concerns regarding the limited contributions to the community.
- I am also concerned with the hyper-parameter sensitivity across different datasets. It seems that the results in Figure 3 are the overall results. However, I am still concerned with the results for each dataset. Are the hyper-parameter settings (e.g. $\tau=3.0$) suitable for all datasets? Considering the generality of the proposed method, I suggest the authors report more detailed results and analyses.

**Questions:**

See weaknesses.

---

> ### Author Response · Authors · 2024-11-21
> **Rebuttal by Authors**
>
> **Response to Complexity Concern (W1):**
>
> Thank you for your comments.
> We will first clarify that: **CARPRT does NOT introduce significant computational overhead compared to ZPE**.
> Please see the complexity analysis below for detailed reasons.
>
> The computational complexity of reweighting methods (including class-wise, prompt-wise, and class-aware prompt reweighting) arises from three main sources: embedding calculations using CLIP, similarity score computations, and weight matrix calculation. We note that the **embedding calculation is the most computationally expensive part**, as each image and all prompt-class combinations must pass through the image and text encoder.
> *This computation is shared by all prompt reweighting methods*.
> We denote the complexity of the text (image) encoder pass as $\Delta_T$ ($\Delta_I$). Thus, the overall complexity of CARPRT is:
>
> $O(m\Delta_I + nC\Delta_T + mnC + nC) \approx O(m\Delta_I + nC\Delta_T),$
>
> and for prompt-wise reweighting:
>
> $O(m\Delta_I + nC\Delta_T + mnC + n) \approx O(m\Delta_I + nC\Delta_T).$
>
> We note that $\Delta$ (encoder complexity) is significantly larger than $m$ due to the multi-layered Transformer architecture, making encoder computations the **dominant cost**. Furthermore, $m$ (the number of images) is much larger than $n$ (the number of prompts) and $C$ (the number of classes), underscoring that the main burden lies in the encoder passes rather than subsequent operations.
>
> In conclusion, CARPRT matches ZPE in computational efficiency while delivering better results, as verified across diverse datasets. We will include this complexity analysis in the revision to clarify this point.

---

> ### Author Response · Authors · 2024-11-21
>
> **Response to Limited Performance Gains (W2):**
>
> Thank you for your comments. In our paper (Section 5.2 and 6), we do have noted that this limited improvement may be associated with the quality of the prompt templates, as well as the intrinsic difficulty and task complexity of ImageNet classification tasks.
>
> In response, we conducted additional experiments to investigate the impact of template quality more comprehensively.
> We attempted to generate **two new** template pools (Pool1 and Pool2) using LLMs.
> The details and results are summarized below:
> - **Pool0**: Original templates (as reported in the paper).
> - **Pool1**: We used [Claude 3.5](https://www.anthropic.com/news/claude-3-5-sonnet) to generate 300 templates by introducing the ImageNet’s label space as context, consisting of 100 templates each in the following formats: *"A photo of a {}, a type of XXX."*, *"A XXX photo of a {}."* and *""A XXX of a {}."*
> - **Pool2**: Using [Phi 3.1](https://huggingface.co/bartowski/Phi-3.1-mini-128k-instruct-GGUF), we generated five descriptive templates per ImageNet category, resulting in a total of 5,000 templates.
>
> | Pool   | Method | ImageNet Acc. | Perf. comparison |
> |--------|--------|----------------|------|
> | Pool0  | ZPE    | 67.42          |      |
> |        | CARPRT | 67.81          | +0.39 |
> | Pool1  | ZPE    | 68.18          |      |
> |        | CARPRT | 68.36          | +0.18 |
> | Pool2  | ZPE    | 68.14          |      |
> |        | CARPRT | 68.79          | +0.65 |
>
> **Key observations**.
> Compared with Pool0
> - Pool1 targets more task-specific information (templates are generated with respect to the ImageNet label space), leading to performance gain for both prompt reweighting strategies ZPE and CARPRT.
> - Pool2, as the generated templates contain more class-specific information, CARPRT exhibits greater performance gains. Its class-aware prompt reweighting may have effectively utilized the class-specific descriptive information, outperforming ZPE's class-independent approach.
>
> The results indicate that enhancing template quality does yield performance improvements.
>
>
> **To further substantiate the contribution of our work**,
> We additionally explore whether our method can benefit **broader research communities**.
> We extend our prompt reweighting method to ProDa [1], a *prompt tuning* method that trains multiple invisible text prompts per dataset.
> Since the problem setting transitions from zero-shot to few-shot, our estimation in Eq. (15) evolves from pseudo-labels to ground-truth labels.
> Our combined approach demonstrated enhanced performance across various datasets, as shown in the table below:
>
> | Dataset      | ProDA  | ProDA + CARPRT |
> |--------------|--------|----------------|
> | Caltech101   | 91.3   | **95.4**       |
> | DTD          | **70.1** | 69.6         |
> | EuroSAT      | **84.3** | 83.4         |
> | Aircraft     | 36.6   | **36.9**       |
> | Food101      | 82.4   | **88.1**       |
> | Flower102    | 95.5   | **95.6**       |
> | Pets         | 90.0   | **93.7**       |
> | Cars         | 75.5   | **78.6**       |
> | **Average**  | 78.2   | **80.2**       |
>
> **Key Observations**.
> These results demonstrate that CARPRT effectively complements ProDA, delivering tangible improvements in most datasets and a significant overall boost in the average performance from 78.2 to 80.2.
> By integrating CARPRT with an established prompt engineering technique, we show its versatility and potential to enhance existing prompt engineering frameworks, demonstrating that CARPRT can **not only help with the usage of visible (natural language) prompts but also facilitate methods based on invisible (text embedding) prompts**.
>
> We will also include these results in the revision to emphasize the broader relevance and robustness of our method.
>
> [1] Lu, Y., Wei, F., Lu, W., and Yu, D. "Prompt Distribution Learning." *CVPR*, 2022.

---

> ### Author Response · Authors · 2024-11-21
> **Rebuttal by Authors**
>
> **Response to Hyperparameter Sensitivity (W3):**
>
> Thank you for your thoughtful feedback.
>
> In the zero-shot classification setting, where only test data is available, conventional hyperparameter selection (typically on a validation set) is not feasible due to the absence of training and validation sets. Our goal is to identify hyperparameters that **exhibit robust and consistent performance across diverse datasets**.
>
> To address your concern, we conducted further experiments regarding hyperparameter sensitivity, on three representative datasets: Caltech, DTD, and Oxford Pets.
> As shown in the table below, we experimented with various temperature settings.
> Among these results, it is evident that a temperature of 3.0 provides a balance of performance across the datasets, making it a reasonable choice given the constraints in zero-shot setting.
> | Temperature | Caltech | DTD   | Oxford Pets |
> |------|---------|-------|-------------|
> | 1    | 87.97   | 46.16 | 74.2        |
> | 2    | 91.91   | 47.56 | **82.76**       |
> | 3    | 92.60    | **47.74** | 82.48   |
> | 4    | 92.56   | 47.10  | 81.86       |
> | 5    | **92.72** | 47.08 | 81.74       |
> | 10   | 92.68   | 47.24 | 81.34       |
>
> We recognize the importance of comprehensive hyperparameter sensitivity analysis and will include results for all datasets in the revision.
>
> We hope these additional results help to address your concerns.

---

> ### Author Response · Authors · 2024-11-24
> **Reminder - Discussion Stage Closing Soon - 24 November**
>
> Dear Reviewer rVkh,
>
> We appreciate the time and effort that you have dedicated to reviewing our manuscript.
>
> We have carefully addressed all your queries. Could you kindly spare a moment to review our responses?
>
> Have our responses addressed your major concerns?
>
> If there is anything unclear, we will address it further. We look forward to your feedback.
>
> Best regards,
>
> Authors of  Submission6458

---

> ### Author Response · Authors · 2024-11-25
> **Reminder - Discussion Stage Closing Soon - 25 November**
>
> Dear Reviewer rVkh,
>
> We appreciate the time and effort that you have dedicated to reviewing our manuscript.
>
> We have carefully addressed all your queries. Could you kindly spare a moment to review our responses and let us know if they sufficiently address your concerns?
>
> If anything requires further clarification, we would be glad to provide additional details. We look forward to heading from you.
>
> Best,
>
> Authors of Submission 6458

---

> > ### Comment · Reviewer_rVkh · 2024-11-25
> >
> > Thanks for your response, and sorry for my late reply. I think the complexity analyses make sense. However, for the other two concerns, the additional experiments seem a bit unconvincing. The experiment for new template pools is conducted on ImageNet, while the experiment of ProDA+CARPRT is conducted on the other 8 datasets. Moreover, the experiment of temperature settings is conducted only on 3 datasets, which may cause concerns regarding the generalization ability. The computational complexity of CARPRT is not so high, so I suggest the authors report more thorough results.

---

> ### Author Response · Authors · 2024-11-26
> **Response to follow-up questions**
>
> ### Priority rationale of conducted experiments
>
> > The experiment for new template pools is conducted on ImageNet.
>
> Thank you for your comments. The experiment for new template pools is conducted on ImageNet to specifically address the concern originally raised in W2 about "limited improvement observed in Table 2". Since this issue refers to ImageNet results, we **prioritized** these additional experiments on this dataset to provide a direct response.
>
> > The experiment of ProDA+CARPRT is conducted on the other 8 datasets.
>
> Thank you for raising this concern. The ProDA+CARPRT experiments were carried out on the other 8 datasets because both ProDA  (in their original paper) and our method (in this paper, Table 1) **were evaluated on these datasets**.
> Also, we kindly note that these fine-grained datasets are **commonly used** benchmarks in the prompt tuning literature.
> These additional experiments aim to demonstrate that CARPRT enhances performance **not only for visible prompts** but **also for approaches using invisible prompts**, further demonstrating its potential contribution to *a broader range of communities* beyond zero-shot classification.
>
> ### Ongoing comprehensive experiments
> We acknowledge the need for **more comprehensive evaluations** and thus are conducting additional experiments, including (1) ProDA+CARPRT for ImageNet and (2) generating high-quality template pools for fine-grained datasets, following your queries.
>
> However, it is important to note that, while CARPRT itself introduces negligible computational overhead, generating high-quality template pools using LLMs and training ProDA's learnable prompts require the availability of GPU resources.
> Owing to resource limitations and queuing for the computing server in the university, these experiments are currently underway.
> We will report the results **as soon as they are available**.
>
> ### Selection of temperature hyperparameter
> > Moreover, the experiment of temperature settings is conducted only on 3 datasets, which may cause concerns regarding the generalization ability.
>
> |   Temperature       | Caltech101 | DTD   | EuroSAT | Aircraft | Food101 | Flower102 | Pets  | Cars  | SUN397 | UCF101 | Average |
> |----------------------|------------|-------|---------|----------|---------|-----------|-------|-------|--------|--------|---------|
> | **1**               | 87.97      | 46.16 | 54.07   | 21.13    | 84.42   | 57.97     | 74.05 | 58.10 | 58.29  | 61.07  | 60.32   |
> | **2**               | 91.91      | 47.56 | **56.04** | 22.62    | **85.87** | **68.88**  | **82.76** | 64.35 | 64.85  | 67.91  | 65.28   |
> | **3**               | 92.60      | **47.74** | 55.85   | **22.64**  | 85.78   | 68.58     | 82.48 | 65.02 | **65.49**  | 68.61 | **65.48** |
> | **4**               | 92.56      | 47.10 | 54.62   | 22.56    | 85.68   | 66.42     | 81.86 | 65.26 | 65.48 | **68.69**  | 65.02   |
> | **5**               | **92.72**  | 47.08 | 54.03   | 22.62    | 85.62   | 65.68     | 81.74 | **65.38** | 65.39  | 68.40  | 64.87   |
> | **10**              | 92.68      | 47.24 | 52.59   | 22.30    | 85.47   | 65.05     | 81.34 | 63.23 | 63.23  | 67.53  | 63.86   |
>
> This table summarizes the hyperparameter analysis results **across all fine-grained datasets**. It is important to emphasize that in zero-shot classification, where only test data is available, conventional hyperparameter selection is inherently difficult due to the lack of training or validation data.
> In our paper, a temperature of 3.0 was chosen as it **consistently yields strong results across datasets**.
> While it may not be optimal for each dataset, it provides a practical and generalizable option under the constraints of the zero-shot setting.
>
> We hope this response addresses your concerns and clarifies the priority rationale of our reported supplementary analysis given the limited rebuttal period.
> Further results will be provided as soon as ongoing experiments are completed. Thank you for your constructive feedback.

---

> > ### Author Response · Authors · 2024-11-30
> > **Results of Generating High-Quality Template Pools for Fine-Grained Datasets**
> >
> > Thank you for your patience. We have completed the experiments on generating high-quality template pools for fine-grained datasets, as shown in the table. Similar to ImageNet, the template pools were constructed as follows:
> >
> > - **Pool1:** Using Claude 3.5, we generated 300 templates by introducing ImageNet's label space as context. This consisted of 100 templates each in the following formats:
> >   - "A photo of a {}, a type of XXX."
> >   - "A XXX photo of a {}."
> >   - "A XXX of a {}."
> >
> > - **Pool2:** Using Phi 3.1, we generated **five descriptive templates per class** for each dataset.
> >
> > | Dataset       | Caltech101 | DTD   | EuroSAT | Aircraft | Food101 | Flower102 | Pets   | Cars   | SUN397 | UCF101 | Average | StdDev |
> > |---------------|------------|-------|---------|----------|---------|-----------|--------|--------|--------|--------|---------|--------|
> > | **Pool1**     |            |       |         |          |         |           |        |        |        |        |         |        |
> > | ZPE           | 93.14      | 46.52 | 42.96   | 23.02    | 85.42   | 67.39     | 86.42  | 65.66  | 65.93  | 66.22  | 64.27   |        |
> > | CARPRT        | 93.86      | 46.86 | 43.78   | 23.54    | 85.56   | 69.67     | 87.61  | 65.91  | 66.62  | 66.63  | 65.00   | +0.74    |
> > | **Pool2**     |            |       |         |          |         |           |        |        |        |        |         |        |
> > | ZPE           | 91.15      | 45.41 | 53.68   | 24.76    | 86.32   | 71.15     | 87.86  | 66.16  | 65.13  | 66.74  | 65.84   |        |
> > | CARPRT        | 91.33      | 45.83 | 52.58   | 25.32    | 86.76   | 71.41     | 89.50  | 66.26  | 65.29  | 67.28  | 66.16   | +0.32    |
> >
> >
> > Similar to the ImageNet results, Pool2 generally outperforms Pool0 and Pool1 (with the exception of EuroSAT and DTD). The imporvement of using Pool2 can likely be attributed to LLMs introducing more relevant, class-specific information for these templates. In contrast, the weaker performance on EuroSAT and DTD may stem from their smaller number of classes (10 and 47, respectively), resulting in fewer templates than Pool0 and Pool1.
> >
> > However, unlike the results for ImageNet, we observe that on Pool2, the improvement of CARPRT over ZPE is less significant for fine-grained datasets (e.g., Cars). This might be due to the fact that the templates generated by the LLM for fine-grained datasets exhibit **high structural similarity**. Taking Cars as an example, 314 out of 500 templates are structured as "The {} boasts XXX," whereas for ImageNet, the diversity between classes results in LLMs generating templates with more varied structures.
> >
> > This limitation may be caused by the CLIP text encoder, which inherently assigns similar weights to repeated sentence patterns, focusing more on keywords while overlooking subsequent descriptive details.
> >
> > Regarding **ProDA+CARPRT for ImageNet**, due to the larger size of the dataset and the need for tuning learnable prompts, the computational requirements are significant (this is due to ProDA tuning, not CARPRT itself). We will report the results as soon as they are available.
> >
> > Lastly, we look forward to receiving feedback on our additional temperature hyperparameter results and the results of generating high-quality template pools for fine-grained datasets, and whether they address some of the concerns raised.

---

> ### Author Response · Authors · 2024-12-04
> **[Update 04/Dec/2024] Results Correction!**
>
> Dear reviewer rVkh,
>
> During our attempt to quickly reproduce the implementation of ProDA and report additional results within a very short period, we encountered some minor issues in our implementation that required adjustments. As a result, we have updated our findings. The corrected results are shown below:
>
> | Dataset       | RN50-ep50 ProDA | **RN50-ep50 ProDA+Ours** | VIT16-ep50 ProDA | **VIT16-ep50 ProDA+Ours** |
> |---------------|---------------------------------|--------------------------|----------------------------------|---------------------------|
> | Caltech101    | 90.79                           | **91.25**               | 95.15                           | **95.62**                |
> | DTD           | 69.47                           | **70.05**               | 70.46                           | **71.73**                |
> | EuroSAT       | 78.36                           | **80.54**               | 75.04                           | **78.52**                |
> | Aircraft      | 36.54                           | **36.69**               | 35.87                           | **37.92**                |
> | Food101       | 80.15                           | **80.41**               | 88.12                           | **88.14**                |
> | Flower102     | 94.47                           | **95.25**               | 95.4                            | **96.43**                |
> | Pets          | 89.14                           | **89.53**               | 93.89                           | **93.84**                |
> | Cars          | 74.25                           | **74.37**               | 83.21                           | **83.25**                |
> | SUN397        | 69.36                           | **70.04**               | 76.21                           | **76.79**                |
> | UCF101        | 75.71                           | **76.14**               | 83.74                           | **84.54**                |
> | ImageNet      | 65.06                           | **65.27**               | 71.96                           | **72.07**                |
> | **Average**   | **74.85**                       | **75.41**               | **79.00**                       | **79.90**                |
>
> In the above table:
> - **RN50-ep50** refers to results using **CLIP-ResNet50**, trained for 50 epochs.
> - **VIT16-ep50** refers to results using **CLIP-ViT-B/16**, trained for 50 epochs.
>
> The corrected results demonstrate that CARPRT consistently outperforms ProDA under a fair setting across all datasets. This consistent improvement highlights CARPRT’s robustness and its effectiveness in complementing ProDA to enhance performance in various scenarios.
>
> We sincerely apologize for the oversight and any confusion caused. Thank you for your understanding and continued support.
>
> Additionally, we have, per your request, **completed all the additional experiments**, including those for ImageNet, and included all the corresponding results in the revised manuscript. We hope these new experiments effectively address your concerns.
>
> Best regards,
> Authors of Submission6458

---

### Author Response · Authors · 2024-12-04
**Revision Update and Summary of Changes (part 1/2)**

We appreciate the thorough feedback from all reviewers and have addressed each concern through additional experiments and clarifications.
We outline our key responses and improvements to the manuscript in this summary.

**Responses to Reviewer rVkh**

1. **Computational Complexity.**
- **Concerns**: Does CARPRT introduce computational overhead compared to ZPE?
- **Response**: We clarify that no significant overhead is introduced by CARPRT compared to ZPE. We **conduct a complexity analysis**, showing that both methods rely on the computationally intensive embedding step via the CLIP encoder, which dominates the cost. We also **report actual evaluation elapsed time** for both CARPRT and ZPE.

2. **Limited Performance Improvement**
- **Concerns**: Why are improvements over ZPE limited for ImageNet and for certain fine-grained datasets (Table 1)?
- **Response**: We clarify that limited gains may be attributed to template quality. To address this,
    - We **supplement experiments** by introducing LLM-generated template pools, under which CARPRT consistently outperforms ZPE. We **add a section (Appendix J)** to include the impact of template quality.
    - We also **supplement experiments** that integrate CARPRT with learnable prompts method, demonstrating CARPRT can **facilitate both training/non-training prompt engineering** scenarios, regardless of general datasets (Imagenet) or fine-grained recognition datasets. We **append these results in Appendix K**.

3. **Hyperparameter Selection**
- **Concerns**: Are the hyper-parameter settings suitable for all datasets?
- **Response**: We clarify that a globally well-performing temperature was chosen due to the lack of validation data in the **zero-shot setting**. We **include additional results (Appendix H)** that confirms balanced performance yielded by the selected temperature.

4. **Follow-up Concern: Scope of Supplementary Experiments**
- **Concerns**: Further evaluations are suggested to assess the impact of LLM-generated template pools and CARPRT integration into learnable prompts (ProDA).
- **Response**: We **conduct experiments** with LLM-generated template pools for fine-grained datasets. We **evaluate the performance** of integrating CARPRT with ProDA on **ImageNet**.


**Response to Reviewer 6Qg1**

1. **Performance**
- **Concern**: Limited performance gain on ImageNet.
- **Response**: We clarify that limited gains may be attributed to template quality. We have explored **two ways** to address it:
    - We **supplement experiments** by introducing LLM-generated template pools, under which CARPRT consistently outperforms ZPE. We **add a section (Appendix J)** to include the impact of template quality.
    - We also **supplement experiments** that integrate CARPRT with learnable prompts method, demonstrating CARPRT can **facilitate both training/non-training prompt engineering** scenarios. We **append these results in Appendix K**.

2. **Pseudo-Label Dependency**
- **Concern**: How does pseudo-label quality affect CARPRT's performance?
- **Response**: We **analyze** and **show a moderate correlation** between pseudo-label accuracy and CARPRT's performance exists, and further **introduce iterative refinement** to mitigate performance degradation when the pseudo-label quality is low.

3. **Imbalanced Datasets**
- **Concern**: How does CARPRT perform under class imbalance?
 - **Response**: We **supplement experiments**, evaluating CARPRT under class imbalance by manually constructing an imbalanced CIFAR-10 dataset. CARPRT **consistently outperforms baselines** in imbalanced settings on imbalanced CIFAR-10 (**Appendix I**).

---

> ### Author Response · Authors · 2024-12-04
> **Revision Update and Summary of Changes (part 2/2)**
>
> **Response to Reviewer a8Wx**
>
> 1. **Computational Complexity.**
> - **Concerns**: Does CARPRT introduce computational overhead compared to ZPE?
> - **Response**: We clarify that no significant overhead is introduced by CARPRT compared to ZPE. We **conduct a complexity analysis**, showing that both methods rely on the computationally intensive embedding step via the CLIP encoder, which dominates the cost. We also **report actual evaluation elapsed time** for both CARPRT and ZPE.
>
>
> 2. **Effectiveness with Learned Prompts**
> - **Concern**: Is CARPRT effective for learned prompts?
> - **Response**: We also **supplement experiments** that integrate CARPRT with learnable prompts method (ProDA), demonstrating CARPRT can **facilitate both training/non-training prompt engineering** scenarios, regardless of general datasets (Imagenet) or fine-grained recognition datasets. We **append these results in Appendix K**.
>
> **Response to Reviewer CBgL**
>
> 1. **Probabilistic Formulation and Correspondences**
> - **Concern**: The probabilistic correspondences are not explicitly presented. Can a mathematical derivation be included to assist understanding?
> - **Response**: We **include detailed mathematical derivations in Appendix F**, connecting implementation (Eqs. (13-16)) to the probabilistic framework. We **correct the presentation error in Eq. (13)** to align the theoretical framework with our implementation.
>
> 2. **Robustness to Noisy Pseudo-Labels**
> - **Concern**: Is CARPRT more sensitive to noisy pseudo-labels than ZPE?
> - **Response**: We analyze the pseudo-label accuracy obtained across different datasets and calculated its correlation with CARPRT's performance improvement over ZPE, demonstrating a **weak-to-moderate correlation**.
>
> 3. **Impact of Template Pool Diversity**
> - **Concern**: How does template pool diversity affect CARPRT?
> - **Response**: We **supplement experiments** by introducing LLM-generated template pools with **increased template diversity**, under which CARPRT consistently outperforms ZPE. We **add a section (Appendix J)** to include the impact of template quality.
>
> 4. **Applying Priors in Appendix D**
> - **Concern**: How can the priors discussed in Appendix D be obtained and used in Test-Time Adaptation?
> - **Response**: We **expand Appendix G** to provide practical examples and guidelines for deriving and applying these priors, particularly for Test-Time Adaptation.
>
> 5. **Proposition 3 Clarifications**
>  - **Concern**: In Proposition 3, why do all the three (Ln 880-883) have $y_1$ as their variates? How are Ln 886-889 derived?
> - **Response**: We revise the proof Proposition 3 to include step-by-step derivations in **Apendix E** and clarify the role of $y_1$ and how the related equations are derived.
>
>
> ---
> We believe these responses have comprehensively addressed **all reviewer concerns**, including follow-up issues raised by reviewer rVkh.
> The revision incorporates additional experiments, elaborated theoretical derivations and clarifications.
> The reviewers (6Qg1, a8Wx, CBgL) have not raised further issues and have **indicated scores leaning towards acceptance**.
>
> Once again, we appreciate the reviewers’ time and efforts, which have greatly enriched the quality and clarity of our work.
>
> Best regards,
>
> Authors of submission 6458

---

### Meta-Review · Area_Chair_738q · 2024-12-20

**Metareview:**

This work aims to improve the zero-shot performance of CLIP by leveraging unlabeled data. Specifically, a class-aware reweighting strategy is proposed to refine the ensemble of multiple prompts according to scores of image-prompt pairs. Reviewers are concerned about computational complexity, a marginal improvement compared to baselines, the sensitivity of hyperparameters, the quality of pseudo labels, lack of strong baselines. While the rebuttal addressed some concerns, the major concerns about performance remained. AC notices that some recent work is proposed to improve CLIP by applying transductive methods on unlabeled data [1] as this work while showing a much better performance. Authors are encouraged to incorporate comments and experiments in discussions to polish the work for resubmission.

[1] Intra-Modal Proxy Learning for Zero-Shot Visual Categorization with CLIP. NeurIPS 2024.

**Additional Comments On Reviewer Discussion:**

After discussion, the concern about computational complexity (Reviewer rVkh, a8Wx) was addressed. Reviewer a8wx raised the score to 6. However, the major concerns of Reviewer rVkh were not fully addressed and the reviewer kept the score of 5 after discussion.

---

### Decision · Program_Chairs · 2025-01-22

Reject